

# A new parameterization of the UV irradiance altitude dependence for clear-sky conditions and its application in the on-line UV tool over Northern Eurasia

N. Chubarova[1], Ye. Zhdanova[1], Ye. Nezval[1]

[1]Faculty of Geography, Moscow State University, GSP-1, 119991, Moscow, Russia

*Correspondence to*: Nataly Chubarova (chubarova@geogr.msu.ru)

**Abstract.** A new method for calculating the altitude UV dependence is proposed for different types of biologically active UV radiation (erythemally-weighted, vitamin-D-weighted and cataract-weighted types). We show that for the specified groups of parameters the altitude UV amplification ($A_{UV}$) can be presented as a composite of independent contributions of UV amplification from different factors within a wide range of their changes with mean uncertainty of 1% and standard deviation of 3% compared with the exact model simulations with the same input parameters. The parameterization takes into account for the altitude dependence of molecular number density, ozone content, aerosol and spatial surface albedo. We also provide generalized altitude dependencies of the parameters for evaluating the $A_{UV}$. The resulting comparison of the altitude UV effects using the proposed method shows a good agreement with the accurate 8-stream DISORT model simulations with correlation coefficient *r>0.996*. A satisfactory agreement was also obtained with the experimental UV data in mountain regions. Using this parameterization we analyzed the role of different geophysical parameters in UV variations with altitude. The decrease in molecular number density, especially at high altitudes, and the increase in surface albedo play the most significant role in the UV growth. Typical aerosol and ozone altitude UV effects do not exceed 10-20%. Using the proposed parameterization implemented in the on-line UV tool (http://momsu.ru/uv/) for Northern Eurasia over the PEEX domain we analyzed the altitude UV increase and its possible effects on human health considering different skin types and various open body fraction for January and April conditions in the Alpine region.

**Keywords:** UV radiation, altitude dependence, RT modelling, erythemally-weighted irradiance, vitamin D-weighted irradiance, cataract-weighted irradiance, interactive UV-tool.

## 1. Introduction

Biologically active UV radiation (BAUVR) is an important environmental factor, which significantly affect human health and nature (UNEP, 1998; UNEP, 2011). Enhanced levels of UV radiation lead to different types of skin cancer (basal and squamous cell carcinomas, cutaneous melanoma), to various eye diseases (cataract, photokeratitis, squamous cell carcinoma, ocular melanoma, variety of corneal/conjunctival effects), and to immunosuppression. However, small doses of UV radiation have a positive effect on health through the vitamin D generation (UNEP, 2011).

UV radiation is affected by astronomical factors (solar zenith angle, solar-earth distance), by different atmospheric characteristics (total ozone content, cloudiness, aerosol, optically-effective gases), and by surface albedo (Bais et al., 2007, Bekki et al., 2011). However, the altitude above sea level has also a significant influence on UV radiation


(Bais et al., 2007). There are a lot of studies and special field campaigns in different geographical regions, which
were devoted to the analysis of the altitude UV effect (Bernhard et al., 2008, Blumthaler and Ambach, 1988,
Blumthaler et al., 1994, Blumthaler et al., 1997, Dahlback et al., 2007, Lenoble et al., 2004, Piacentini, et al. 2003,
Pfeifer et al., 2006, Sola et al., 2008, etc.). The UV enhancement at high altitudes is detected not only due to smaller
molecular scattering, but also due to usually observed decreasing in total ozone content and aerosol, and increasing
in surface albedo, which in turn enhances 3D reflection from slopes of mountains covered by snow (Lenoble et al.,
2004). In addition, variation of cloud properties with altitude can also change the level of UV radiation.
The UV records in mountainous areas demonstrate extremely high levels. The highest UV values are observed in
Andes in Bolivia (Pfeifer et al., 2006, Zaratti et al., 2003), where the UV index can be sometimes close to 20. Very
high UV levels were also recorded at high-altitude deserts in Argentina (Piacentini, et al. 2003). In Tibet the UV
index frequently exceeded 15 on clear days and occasionally exceeded 20 on partially cloudy days (Dahlback et al.,
2007). At the European alpine stations in summer conditions the UV indices are often higher than 11 (Hülsen,
2012). In winter, erythemally-weighted irradiance is about 60% higher than that at lower-altitude European sites
(Gröbner et al., 2010). The analysis of erythemal UV doses at different sites in Austria and Switzerland also
demonstrates a significant growth of UV radiation with the altitude (Rieder et al., 2010). In the Arctic the
comparison of summer UV measurements at Summit (3202m a.s.l.) and Barrow (~0 a.s.l.) stations also shows
significant enhancing of about 30-40% in clear sky conditions at the elevated site (Bernhard et al., 2008).
The UV altitude gradients obtained from model calculations vary within the range of 3.5-6%/km in the cloudless
atmosphere if all other parameters (ozone, aerosol, surface albedo) do not change with the altitude (Chubarova and
Zhdanova, 2013). Even smaller values of the estimated UV altitude gradients (3.5%/km) were obtained in conditions
with high surface albedo at both sea level and high altitude, since the larger diffuse component at sea level, to some
extent, compensates the higher direct flux due to a smaller total optical depth at higher altitudes. However, the
experimental UV altitude gradients are often much higher due to the presence of additional altitude changes in the
atmospheric parameters. According to different field campaigns UV altitude gradients vary within 5-11%/km
(Pfeifer et al., 2006, Zaratti et al., 2003, Schmucki, Philipona, 2002), 11-14%/km according to (Sola et al., 2008),
and in some cases can reach 31%/km (Pfeifer et al., 2006). The existence of spectral dependence in absorption
coefficients of ozone as well as in molecular scattering cross sections provides a pronounced spectral character of
the altitude UV effect, which was obtained in many publications (Blumthaler et al., 1994, Sola et al., 2008).
However, the continuous UV records in mountainous area are still very rare due to the complexity of accurate UV
measurements in severe conditions. The accurate results of measurements from different field campaigns devoted to
the evaluation of altitude UV effects shown in (Bernhard et al., 2008, Blumthaler and Ambach, 1988, Blumthaler et
al., 1994, Blumthaler et al., 1997, Dahlback et al., 2007, Lenoble et al., 2004, Piacentini, et al. 2003, Pfeifer et al.,
2006, Sola et al., 2008, Zaratti et al., 2003) provide precise, however, local character of this phenomenon, which
results in various altitude UV gradients.
At the same time, the accurate RT (Radiative Transfer) model simulations (Liou, 2010) are very time consuming
and can not be used in different on-line tools or other applications. There are also a lot of UV model assessments for
the past and future UV climate scenarios but usually they are given with the coarse spatial resolution, which does
not allow a user to obtain the accurate estimates over the particular mountainous location.
In addition, the limiting factor of the UV calculation accuracy is the uncertainty of input geophysical parameters,
which significantly increases at high altitudes. Hence, another task was to obtain some generalized dependencies of
the input parameters using the available data sources.





The objective of this paper is to provide the accurate parameterization for different types of biologically active
radiation for the estimation of UV level at different altitudes taking into account the generalized altitude dependence
of different geophysical parameters. Using the proposed parameterization we will also discuss the consequence of
the enhanced UV level at high altitudes for human health using the classification of UV resources via a specially
developed on-line interactive UV tool.
**2. Materials and methods**
In order to account for different effects of UV radiation on human health we analyze three types of BAUVR:
erythemally-weighted, vitamin D-weighted, and cataract-weighted irradiances, which are calculated using the
following equation:
$$Q_{bio} = \int_{280}^{400} Q_\lambda F_\lambda d\lambda,$$
(1)

where $Q_\lambda$ is the spectral flux density, $F_\lambda$ is the respective biological action spectrum.
We used erythemal action spectrum according to CIE (1998), vitamin D spectrum - according to CIE (2006), and
cataract-weighted spectrum according to Oriowo et al. (2001). Various types of BAUVR action spectrum have
different efficiency within the UV range. For their characterization we used the effective wavelengths, which are
calculated as follows:
$\lambda_{eff} = \int Q_\lambda \lambda \, d\lambda / \int Q_\lambda d\lambda$
(2)

According to our estimates, for example, at high solar elevation (h=60°) and for the variety of other parameters
(total ozone, aerosol and surface albedo) the effective wavelength for erythemally-weighted irradiance ($Q_{ery}$) is
~317 nm, for cataract-weighted irradiance ($Q_{eye}$) - ~313 nm, and for vitamin D-weighted irradiance ($Q_{vitD}$) -
~308 nm. These changes in effective wavelengths for various BAUVR types indicate their different sensitivity to the
ozone absorption, molecular scattering and aerosol attenuation, which vary dramatically within this spectral range,
and, as a result, explain different BAUVR responses to the changes in these geophysical parameters.
All the simulations were fulfilled using one dimensional radiative TUV (Tropospheric Ultraviolet-Visible) model
with 8-stream DISORT RT method (Madronich and Flocke, 1997) and 1 nm spectral resolution. The uncertainty of
the RT method is less than 1% (Liou, 2010). Badosa et al.(2007) showed a good agreement between the
experimental spectral data in different geographical regions and simulated results using this RT method if the input
atmospheric parameters were known.
Several experimental datasets were used. For obtaining the generalized altitude dependence of aerosol optical depth
(AOD) we used the data of sun/sky CIMEL photometers from different AERONET sites located at different heights
above sea level (Holben et al., 1998)). These data account for the near-ground emission sources of the aerosol at
various altitude in the aerosol column content. The estimated uncertainty for aerosol optical depth in UV spectral
region is about 0.02. The uncertainty for single scattering albedo is about 0.03 at AOD440>0.4 and the uncertainty
for all other inversion parameters is not higher than 10% (Holben et al., 2006). In addition, the dataset of historical
Moscow State University complex field campaigns over mountainous areas at Pamir (38- 40.5° N, 73-74° E
H=1.0÷3.9 km), and Tyan'Shan' (43°N, 77°E, H=3.47km) was applied in the analysis (Belinski et al., 1968). It





includes the data records of total ozone content and aerosol optical depth at 330 nm, which had been measured with
the help of M-83 filter ozonometer, and UV irradiance less 320 nm – by the UVM-5 instrument calibrated against
the spectroradiometer BSQM (the Boyko's Solar Quartz Monochromator) described in (Belinski et al., 1968). The
description of the BSQM and the details of the calibration were also discussed in Chubarova and Nezval' (2000).
The uncertainties of UV measurements less 320nm due to the calibration procedure were considered to be about
10% (Belinsky et al., 1968). However, to avoid the calibration errors only relative measurements were used in this
study. The residual uncertainty due to possible existence of slight variation in spectral response of the instrument
and their temperature dependence was estimated to be about 5-7%. The obscurity of the horizon at all sites was less
than 10°. The field campaigns were carried out during summer periods, when no snow was detected at the surface.
The snow covered mountainous peaks were only observed at Tyan'Shan' at relatively large distance of more than 30
km from the site.
We also used the LIVAS database (Lidar Climatology of Vertical Aerosol Structure for Space-Based Lidar
Simulation Studies, http://lidar.space.noa.gr:8080/livas/). This is a 3-dimensional global aerosol climatology based
on satellite lidar CALIPSO observations at 532 and 1064 nm, EARLINET ground-based measurements and a
combination of input data from AERONET, aerosol models, etc. The final LIVAS climatology includes 4-year
(2008 – 2011) time-averaged 1×1° global fields (Amiridis, et al., 2015). We used the annual aerosol extinction
profiles at 355 nm for calculating aerosol optical depth over various points at different altitudes in the Alpine and
the Caucasian mountainous regions in Europe and over the high-elevated regions in Asia. It should be mentioned
that the LIVAS averages all Calipso overpasses over a 1x1° cell and characterizes only the mean altitude within the
cell. This provides some additional uncertainties in its aerosol extinction altitude dependence evaluation. On
average, according to (Amiridis et al., 2015) the absolute difference in LIVAS AOD is within 0.1 agreement with
AERONET AOD values in UV and visible region of spectrum.
In addition, we estimated UV resources at different altitudes according to the approach given in Chubarova and
Zhdanova (2013), which has been developed on the base of international classification of UV index (Vanichek et
al., 2000) and the vitamin D threshold following the recommendations given in CIE (2006). According to this
approach we defined *noon UV deficiency* and *100% UV deficiency categories*, when UV dose is smaller than the
vitamin D threshold, and it is not possible to receive vitamin D respectively during solar noon hour, and throughout
the whole day. The *UV optimum* category is determined when the UV dose does not exceed erythema threshold but
it is possible to receive UV dose, necessary for vitamin D at noon hour. Several subclasses of *UV excess* are
attributed to the thresholds depending on the standard UV index categories: *moderate UV excess* class, which relates
to moderate category of hourly UV index, *high UV excess, very high UV excess*, and *extremely high UV* excess
category. Currently, in the assessment of UV resources we do not take into account for the eye damage UV effects,
since there is no reliable regulation on the UV threshold for this type of BAUVR.
**3. Results**
**3.1. The general description of the approach**
It is widely known that "the solution of the radiative transfer equation is possible to derive by numerous solution
methods and techniques" (Liou, 2010). However, the accurate RT methods usually require a lot of computer time
and can not be used in several applications. The simulated intensity and UV flux density (or irradiance) has a
complicated non-linear dependence on many geophysical parameters, however, our numerous simulations of UV



irradiance using the accurate 8-stream discrete ordinate RT method show that within a variety of geophysical
parameters one can obtain the parameterized altitude correction by taking into account for the quasi-independent
terms driven by different geophysical factors. Some of them are independent due to different vertical profiles (for
example, ozone maximum in the stratosphere compared with aerosol and molecular maximum in the troposphere).
Some of them are dependent (for example, surface albedo UV effects depend on molecular and aerosol loading),
but, as we show later, this factor can be also considered as a one joint term.
Using this assumption, we propose a parameterization, where biologically active UV irradiance at the altitude $H$
($Q_{bio}(H)$) can be estimated from $Q_{bio}$ at zero altitude ($H=0$ km a.s.l.) with an independent account for the terms,
which are affected by different geophysical factors:
$Q_{bio}(M_H,X_H,AOD_H,S_H)=Q_{bio}(M_0,X_0,\ AOD_0,S_0)\cdot A_M A_X A_{AOD} A_{S(M,AOD,cloud)}$,     (3)
where $A_M$, $A_X$, $A_{AOD}$ are the UV amplifications, respectively, due to the altitude decrease in molecular number
density ($M$), ozone ($X$), and aerosol optical depth ($AOD$). $A_{S(M,AOD,cloud)}$ is the UV amplification due to the increase in
surface albedo $S$, which is typically observed with the altitude. This characteristic is also a function of a change in
molecular number density, aerosol and cloud characteristics with height due to the processes of multiple scattering.
Further, only the effects in the cloudless atmosphere are considered. The total UV amplification ($A_{uv}$) with altitude $H$
can be rewritten from Eq.(3) as:
$A_{UV}=A_M A_X A_{AOD} A_S=\dfrac{Qbio(M_H,X_H,AOD_H,S_H)}{Qbio(M_0,X_0,AOD_0,S_0)}$     (4)
Let us consider separately the effects of different factors on UV irradiance at high altitudes. We specify them by
using the accurate RT model simulations, different empirical datasets or by applying the important characteristics
from different publications.
The possibility of this approach was tested directly by the accurate modelling for a variety of conditions at different
solar elevations. The model simulations were made for the altitude changes from zero to 5 km with the variations of
aerosol optical depth at 340nm within $AOD_{340}\sim0.0$-$0.4$, variations in total ozone from 350 to 250 DU, and surface
albedo changes from zero to $S=0.9$ at different altitudes. As the input aerosol parameters within UV spectral region
we used single scattering albedo $SSA=0.96$, factor of asymmetry $g=0.72$, and Angstrom exponent of $\alpha=1.0$, which
are close to the aerosol background characteristics in Europe (Chubarova, 2009). We compared the $A_{UV}$ values
calculated as a multiplication composite of different separate parameters ($A_M$, $A_X$, $A_{AOD}$, and $A_S$) according to Eq.(4)
with the $A_{UV}$ values, which were estimated as a ratio of direct simulations of BAUVR at the altitude $H=5$ km and at
zero ground level. The results of the comparisons are shown in Fig.1. One can see a good agreement between the
$A_{UV}$ values obtained using multiplication of $A_M A_X A_{AOD} A_S$ and the $A_{UV}$ values from direct estimations of BAUVR.
The correlation for all BAUVR types is higher than 0.99 with the mean relative difference of -1±3% compared with
the exact model simulations with the same input parameters. The slight variations in aerosol parameters (within
10%) does not change the obtained results.
**3.2. Molecular UV amplification with the altitude**
A decrease of atmospheric pressure, or molecular number density, with the height is a well-known factor of UV
amplification. According to the 8-stream DISORT model simulations we found that the BAUVR dependence with
the altitude has a linear change in the molecular atmosphere, which is clearly seen in Fig.2. Hence, for its
characterization we can apply a simple gradient approach.





For evaluating the UV amplification due to molecular effects the following expression is used:
$A_M = \dfrac{Qbio(M_H,X_0,AOD_0,S_0)}{Qbio(M_0,X_0,AOD_0,S_0)} = 1 + 0.01 G_{bio,M}(S=0) \Delta H$   (5)
where $G_{bio,M}$ is the relative molecular gradient, in %/km, $\Delta H$ is the difference in the altitudes, in km. Note, that all
other parameters do not change with the height.
The estimated relative molecular gradients for different types of BAUVR for various conditions are shown in Table
1. At solar elevation $h=10°$ there is a decrease in the $G_{bio,M}$ for different BAUVR and, especially, for vitamin-D
irradiance due to its smaller effective wavelength and the effects of stronger ozone absorption, which is increased at
higher ozone content (X=500 DU). However, for solar elevation higher than 20° the sensitivity of the $G_{bio,M}$ values is
around 6-7%/km and does not significantly change with variations in $h$ and $X$.
As an example, at the altitude of 5km and at high solar elevation the molecular UV amplification according to Eq.
(5) lies within ~1.26-1.38 depending on the type of BAUVR (see Table 1), which is in accordance with the accurate
model simulations. However, at $h=10°$ the UV amplification for erythemally-weighted and cataract-weighted
irradiances is about 1.18-1.23, while for vitamin D-weighted irradiance $A_M$ is only 1.04-1.09 depending on ozone
content. The maximum UV amplification at the highest peak (m. Everest, $H$=8.848 km) due to changes only in
molecular scattering reaches 1.53-1.68 at high solar elevation depending on the type of BAUVR.
**3.3. Ozone UV amplification with the altitude**
In order to account for the ozone decrease with the altitude we apply the existing linear dependence between UV
radiation and total ozone $X$ in log-log scale. This approach was used in the definition of the Radiation Amplification
Factor (*RAF*) by Booth and Madronich (1994). As a result, the following equation can be written:
$log(Q_{bio}) = RAF(Q_{bio,h}) \, log(X_i) + C$,   (6)
where $h$ is the solar elevation, $C$ is the constant.
The *RAF* values vary for different types of BAUVR. For example, at high solar elevation Radiation Amplification
Factor for erythemally-weighted irradiance $RAF_{Qery} = 1.2$, for vitamin D-weighted irradiance - $RAF_{QvitD}=1.4$, for
cataract weighted irradiance - $RAF_{Qeye}=1.1$ (UNEP, 2011). However, we should take into account the RAF
dependence on solar elevation $h$ due to the relative changes in solar spectrum with $h$. Using the results of accurate
RT modelling we have obtained *RAF* dependencies on $h$ for different types of BAUVR:
$RAF_{Qery}(h) = -1.10E\text{-}04 \pm 1.49E\text{-}5 \; h^2 + 1.57E\text{-}02 \pm 1.53E\text{-}3 \; h + 0.665 \pm 0.0333$   (7)
$R^2 = 0.98$
$RAF_{QvitD}(h) = 0.000166 \pm 0.00001 \; h^2 - 0.0277 \pm 0.0011 \; h + 2.5121 \pm 0.0233$   (8)
$R^2 = 0.997$
$RAF_{Qeye}(h) = 1.43E\text{-}6 \pm 1.0E\text{-}6 h^3 - 0.000202 \pm 0.000066 \; h^2 + 0.00483 \pm 0.0029 \; h + 1.297 \pm 0.035$   (9)
$R^2 = 0.98$
where $R^2$ – is the determination coefficient. The standard error estimates of the coefficients in the equations are
given at P=95%.
Note, that similar approach for accounting the RAF solar angle dependence was proposed in Herman (2010) with
higher power degree.





As a result, the BAUVR at the altitude $H(Q_{bioH})$ with the correction on ozone content can be written as follows:
$$Q_{bioH}=Q_{bio0}(X_0/X_H)^{RAF(Qbio,h)} \tag{10}$$
From Eq.(10) we can obtain the altitude UV amplification due to ozone using the altitude ozone gradient $G_X$
(DU/km):
$$A_{X=}\frac{Qbio(M_0,X_H,AOD_0,A_0)}{Qbio(M_0,X_0,AOD_0,A_0)} = \left(\frac{X0}{X0-G_X*\Delta H}\right)^{RAF(Qbio,h)} \tag{11}$$
We propose to apply the typical ozone altitude gradient $G_X$, which absolute value is about 3.5 DU/km according to
monthly averaged ozone soundings measurements in Germany and observations in Bolivia (Reuder and Koepke,
2005; Pfeifer et al., 2006).
As an example, if we take into account only for this typical ozone decrease with the altitude, the UV enhancement at
5 km will be about $A_X$ ~1.06-1.11 while at the highest peak (m. Everest) $A_X$ will reach 1.11-1.22 at high solar
elevations depending on the BAUVR type and initial ozone content at zero altitude within $X$=250-350 DU.
**3.4. Aerosol UV amplification with the altitude**
Aerosols can significantly change their characteristics with the altitude, affecting the level of BAUVR. Due to
variations in size distribution and optical properties aerosol may have different radiative properties (aerosol optical
depth, single scattering albedo, and phase function). One of the most important aerosol characteristics affecting UV
radiation is aerosol optical depth.
For accounting the aerosol effect on UV attenuation we propose to apply the equation given in (Chubarova, 2009):
$$Q_{bio}(AOD_{340})=Q*_{bio(AOD=0)}(1+AOD_{340} B) \tag{12}$$
where $B=(0.42m+0.93) SSA-(0.49m+0.97)$, $Q*_{bio}$ is the BAUVR in aerosol free conditions, $m$ is the air mass, $SSA$ is
the single scattering albedo.
The Eq. (12) was obtained from the accurate model simulations for the conditions with low surface albedo ($S$=0.02),
which is typical for grass. (Here and further we consider AOD at wavelength 340nm ($AOD_{340}$), since this
wavelength corresponds to the standard UV channel in CIMEL sun/sky photometer, which is used in AERONET).
The coefficients were obtained according to model simulations for $0<AOD_{340}<0.8$, single scattering albedo
($0.8<SSA<1$), and airmass $m\sim sinh^{-1}$ ($m\leq 2$), Angstrom exponent $\alpha$~1. Since the radiative effects of the existing AOD
spectral dependence are relatively small within the UV-B spectral range we consider the same coefficients for
different types of BAUVR.
Assuming that single scattering albedo and factor of asymmetry do not change with the altitude, we evaluated the
UV amplification with the altitude due to aerosol optical depth. Using Eq.(12) the equation for $A_{AOD340}$ can be
written as follows:
$$A_{AOD340 =}\frac{Qbio(M_0,X_0,AOD_H,A_0)}{Qbio(M_0,X_0,AOD_0,A_0)} = \frac{1+AOD_{340,H}B}{1+AOD_{340,0}B} \tag{13}$$
In some conditions single scattering albedo and asymmetry factor may have the altitude dependence (see, for
example, the results of aircraft measurements in (Panchenko et al., 2012)). However, the uncertainty of neglecting
the altitude changes in single scattering albedo significantly decreases at small AOD observed at high altitudes. We
should also note that only the altitude changes in aerosol optical depth are usually taken into account in the standard
tropospheric aerosol models (WMO, 1986).





The aerosol optical depth at the altitude $H(AOD_{340,H})$ can be evaluated using the following expression:
$AOD_{340,H}=AOD_{340,0}f_{AOD}(H),$                     (14)
where $f_{AOD}(H)$ is the altitude dependence of aerosol optical depth.
There are a lot of model aerosol profiles for the free atmosphere conditions (see, for example, widely used aerosol
models in WMO, (1986)). However, these profiles can not be applied for high-elevated locations, which are usually
characterized by a significant emission of primary aerosols or their precursors from nearby surface even in
background conditions. To account for this kind of altitude AOD dependence we used different ground-based and
satellite measurements described in the Section 2. Since our objective was to obtain the generalized aerosol altitude
dependencies we used the monthly mean *AOD* data from different archives over different geographical regions in
Eurasia. The dependence of aerosol optical depth as a function of altitude is shown in Fig.3. Highly variable $AOD_{340}$
values at different altitudes may be roughly combined in two groups, which are characterized by different rates of
aerosol altitude decrease. Hence, in our parameterization we propose to distinguish these two types of the altitude
aerosol dependence. The first one is characterized by a very strong aerosol optical depth decrease with the altitude.
It was obtained mostly from the data of European AERONET sites in the Alpine zone as well as from several Asian
sites in the sharp-peak mountainous areas. This dependence was also confirmed by the LIVAS dataset
measurements over the same areas.
The second one is characterized by a much more gradual altitude $AOD_{340}$ decrease observed over flat elevated Asian
regions. The main reason of such a character is the existence of the additional aerosol emission sources (i.e. loess,
mineral aerosol) from the vast areas of deserts and semi-deserts elevated over sea level of up to 3-4 kilometers.
In addition, Fig.3 demonstrates the $AOD_{340}$ dependence on altitude according to the data obtained during Moscow
State University field campaigns at the high-altitude plateau at Pamir and Tyan'-Shan' mountainous regions in
Central Asia. We can see its satisfactory agreement with the second type of $f_{AOD}(H)$ obtained from the AERONET
and LIVAS dataset.
The first, Alpine – like type, can be parameterized as:
$f_{AOD}(H)=H^{-1.65}, R^2=0.4$                     (15)
The Alpine type aerosol altitude dependence was firstly obtained for the simulation of the UV climatology over
Europe (COST 726, 2010). However, the coefficients have been re-affirmed using more statistics.
The second, so-called Asian type, can be obtained using the equation according to the Moscow State University
expedition dataset in Asian region. It is characterized by much flat dependence with the altitude:
$f_{AOD}(H)= exp(-0.26H), R^2=0.8$                     (16)
The proposed dependencies can be considered as the two classes with different altitude aerosol decreasing rates.
Both dependencies are accounted for the altitudes higher than 1 km, since our analysis of AERONET dataset has
revealed the absence of the aerosol altitude dependence at the heights below 1 km due to prevailing the effects of
different aerosol sources or their precursors there. However, over the particular location the altitude *AOD*
dependence within the first kilometre can be found, of course.
We should note that the proposed altitude *AOD* dependencies according to (15) and (16) are considered only as a
first proxy for the most sharp and flat altitude dependencies. For a particular location and specific geographical
conditions the AOD altitude dependence can be different. However, a user may easily substitute them in the
proposed parameterization.



Although the *AOD* altitude dependence is pronounced, its influence on UV amplification highly depends on initial
aerosol conditions at $H$=0 km, the type of the altitude profile, and solar elevation (see Eq. (12)). For example, for the
Alpine type aerosol altitude profile the UV amplification at $H$=5 km is about $A_{AOD}$=1.05-1.10 and does not exceed
*1.11* at $H$=8.848 km for typical aerosol at $H$=0 km ($AOD_{340}$=0.36). However, for the polluted conditions
characterized by $AOD_{340}$=0.8 at $H$=0 km, the altitude UV amplification at $H$=5 km is about $A_{AOD}$~1.12-1.27
depending mainly on solar elevation. Note, that at $H$=8.848 km the effect is almost the same ($A_{AOD}$~1.16-1.29). This
will be further discussed below.
**3.5. UV amplification due to changes in surface albedo with the altitude**
The increase in surface albedo is one of the important factors, which is necessary to account for the effective
calculations of BAUVR at high altitudes. Due to significant negative temperature gradients, the snow with high
surface albedo can be observed even in summer conditions at high altitudes instead of vegetation with low UV
albedo of about $S$=0.02-0.05 (Feister and Grewe, 1995). Fig.4 demonstrates the enhancement in erythemally-
weighted irradiance due to the increase in surface albedo according to different experimental studies and the results
of one-dimensional model simulations. One can see the UV increase of around 20% at the effective surface albedo
close to $S$=0.5 (Simic et al., 2011, Huber et al., 2004, Smolskaia et al., 2003). On the average, there is an agreement
between the calculation of UV amplification by 1D model and the measurements at different mountainous regions
up to effective surface albedo of $S$~0.5. However, the accurate comparison of UV measurements with 3D model
(Diemoz and Mayer, 2007) shows the additional snow effect of about ±1UV index value due to the account of
overall interactions between radiation and different surfaces. The comparisons of UV spectral actinic flux
measurements with 1D and 3D model simulations demonstrate the similar range of uncertainties of these models,
however, 3D model gives, of course, more realistic view of the UV field in mountains since the topography and the
obstruction of the horizon are taken into account (Wagner et al., 2011). However, currently we do not consider 3D
effects in our parameterization. Due to small UV albedo over snow free surfaces this factor is negligible in summer,
while in winter the value of the effective surface albedo in mountainous area can be very high and significantly
depends on tree line location.
To account for surface albedo effects we followed the results obtained in different papers (Green, et al., 1980,
Chubarova, 1994), where the effects of multiple scattering were accounted using geometric progression approach.
The same approach with a detailed physical analysis was used in (Lenoble, 1998). Following these publications we
propose to calculate biologically active UV radiation in conditions with surface albedo $S$ as follows:
$$Qbio_S = Qbio_{S=0} \frac{1}{1-r_{bio}S}$$ (17)
where $r_{bio}$ is the coefficient, which characterizes the maximum relative change in $Q_{bio}$ due to multiple scattering for
surface albedo variations from 0 to 1. According to (Lenoble, 1998) $r_{bio}$ is determined as the atmospheric reflectance
illuminated on its lower boundary. Note, that surface albedo $S$ characterizes the reflecting properties at ground at the
considered altitude $H$.
The application of the equation (16) to $H=0$ km and to the altitude $H$ allows us to obtain the following expression
for $Q_{bio}$ at $H$ with surface albedo $S_H$: using the known $Q_{bio}$ at the altitude $H$=0 km with surface albedo $S_0$:
$$Q_{bio_{S_H}}(H) = Q_{bio_{S_0}}(H=0) \frac{Q_{bio_{S=0}}(H)}{Q_{bio_{S=0}}(H=0)} \frac{1-r_{bio}(H=0)S_0}{1-r_{bio}(H)S_H}$$ (18)
This equation can be rewritten in the following way:




$$\frac{Q_{bio S_H}(H)}{Q_{bio S_0}(H=0)} = \frac{Q_{bio S=0}(H)}{Q_{bio S=0}(H=0)} \frac{1 - r_{bio}(H=0)S_0}{1 - r_{bio}(H)S_H}$$ (19)
One can see that in Eq. (19) the left side of the equation $\frac{Q_{bio S_H}(H)}{Q_{bio S_0}(H=0)}$ is the total UV amplification $A_{UV}$ defined in
equation (4); the first term ($\frac{Q_{bio S=0}(H)}{Q_{bio S=0}(H=0)}$) at the right side of the equation characterizes the total UV amplification in
conditions with $S=0$, while surface albedo effects are accounted only in the last term. Hence, we can write the UV
amplification due to the effects of surface albedo as follows:
$$A_S = \frac{1 - r_{bio}(H=0)S_0}{1 - r_{bio}(H)S_H}$$ (20)
According to the model estimations the value $r_{bio}$ in clear sky conditions has a relatively small dependence on
altitude, which appears due to a decrease mainly in molecular and aerosol loading and can be easily parameterized
by a simple regression as follows:
$r_{bio}(H) = b\,H + c,$ (21)
where the coefficients $b$ and $c$ are given in Table 2 for different types of BAUVR. They were obtained for a variety
of solar elevation and ozone content taking into account for the altitude changes in molecular scattering as well as
for altitude dependence of aerosol optical depth $f_{AOD}(H)$. The $r_{bio(H)}$ mainly depends on molecular content and
aerosol properties, and slightly decreases with the altitude due to reducing in multiple scattering effects with the
decrease in molecular and aerosol loading.
As a result, the UV amplification due to the increase in surface albedo at the altitude $H$ strongly depends on
scattering processes and also decreases with the altitude. Fig.5 shows the maximum $A_S$ effect due to the changes in $S$
from $S=0$ at zero level to $S=1$ at the altitude $H$ for the different types of BAUVR. The $A_S$ decreases with the altitude
from more than 1.6 at $H=0$ km to about 1.2 at $H=8.848$ km due to the decrease in $r_{bio}$.
**3.6. Validation**
Using the generalized parameterizations for different geophysical parameters obtained in the previous sections we
can estimate the total UV amplification $A_{UV}$ with the altitude from Eq.(4). The results of the validation of the
proposed method with these altitude parameter dependencies against the accurate RT simulations are shown in
Fig.6. One can see a close correlation (r>0.996) between the $A_{UV}$ values obtained by the proposed method and the
accurate RT simulations using the 8-stream DISORT method within the changes in altitude from H=0 km to 8 km,
in solar elevation from 20 to 50°, in surface albedo from $S=0$ to $S=0.9$, in ozone from 250 DU to 350 DU at H=0 km,
and in AOD$_{340}$ from 0.2 to 0.4 at H=0 km. Different altitude aerosol profiles were also considered. Validation was
made for all three types of BAUVR. Overall, the average bias is 0±0.2% for erythemally-weighted irradiance, and
1±0.2% - for other types of BAUVR. The maximum difference between the $A_{UV}$ calculated by the proposed method
and by the accurate model simulations does not exceed 6% at the highest elevation (*H=8* km) at low ozone content.
The comparisons of the total UV amplification according to the proposed method with the total $A_{UV}$ obtained from
the experimental dataset as a function of altitude are shown in Fig.7. The experimental data were taken from the
dataset of Moscow State University mountainous field campaigns, which was described in the Section 2. After
accounting for the molecular, aerosol, and ozone altitude dependence the simulated UV amplification is in
satisfactory agreement with the obtained experimental results.



**4.Discussion**
The total altitude amplification of biologically active UV irradiance $A_{UV}$ as a function of altitude is shown in Fig.8
for a variety of atmospheric conditions at surface albedo $S$=0 and $S$=0.9 and high solar elevation $h$=50°. One can see
a distinct altitude difference obtained for different types of BAUVR with larger increase for vitamin D-weighted
irradiance due to its higher sensitivity to ozone content. The difference in $A_{UV}$ for various BAUVR can reach 15-
20% at high altitudes. The effects of surface albedo on $A_{UV}$ can be seen if compare the results shown in Fig 7a and
Fig.8b. One can see the 2-2.5 times UV increase due to high surface albedo at high altitudes, which is again more
pronounced for vitamin D-weighted radiation with smaller effective wavelength and, hence, more effective multiple
scattering than that for the other types of BAUVR. Larger $A_{UV}$ values are also observed in conditions with smaller
ozone amount for all three types of BAUVR for both zero and high surface albedo conditions. High surface albedo
$S$=0.9 provides a significant increase even at zero level, which is similar to the $A_{UV}$ increase due to altitude change
of 6 km. It is clearly seen that typical aerosol and ozone does not play a vital role in $A_{UV}$. However, for all types of
BAUVR the increase of slightly absorbing aerosol (from $AOD_{340}$=0.2 to $AOD_{340}$=0.4) provides a noticeable $A_{UV}$
growth in conditions with relatively small ozone content due to enhancement of multiple scattering (see Fig.8).
The $A_{UV}$ values are smaller at solar elevation h<20° for all types of BAUVR mainly due to decreasing in $G_{bio\_M}$ (see
the coefficients in Table 1). For example, at $H$=8 km the UV altitude amplification for vitamin D-weighted radiation
is about $A_{UV}$ =1.77 at $h$=10° compared with $A_{UV}$ =2.0 at $h$=50° at $X$=250 DU and $AOD_{340}$=0.4.
Let us consider the conditions, which are characterized by the most pronounced UV amplification with the altitude -
the conditions with high aerosol loading $AOD340$=0.8, low ozone content $X$=250DU at $H$=0 km, and high solar
elevation h=50°. In addition, we consider the abrupt increase in surface albedo at $H$=2 km from $S$~0 to $S$=0.95,
which can be possible due to location of tree line there and pure snow above. The altitude UV amplification due to
these input parameters according to the proposed $A_{UV}$ parameterizationis shown in Fig.9. The separate effects of
different factors can be seen in Fig. 9a and their total effects on different BAUVR types are shown in Fig.9b. One
can see a different role of geophysical factors at different altitudes: the prevalence of molecular scattering especially
at high altitudes while the extremely high surface albedo may play the most important role at the altitudes of its
abrupt increase (in our case - $H$=2km, $A_S$=1.48). However, in our example the UV amplification due to surface
albedo decreases at the altitude higher than 2 km because of the reduction in multiple scattering. The UV altitude
amplification due to aerosol is more distinct and reaches 1.1-1.2 at high altitudes if there is a strong aerosol pollution
$AOD340$=0.8 at $H$=0 km. It is more pronounced for the Alpine-type AOD altitude dependence and in our example at
$H$=2 km it can be even higher than the $A_M$ value (see Fig.9). The effects of ozone in UV amplification do not exceed
1.1-1.20 at high altitudes depending on BAUVR type. We would like to emphasize that Fig.9 is only the illustration
of the application of the proposed $A_{UV}$ parameterization  for a given parameters altitude variations.
We implemented the proposed UV altitude parameterization in the developed on-line UV tool http://momsu.ru/uv/,
which had been developed for the simulation of erythemally-weighted irradiance and the UV resources over
Northern Eurasia (the PEEX domain) at $H$=0 km (Chubarova and Zhdanova, 2013). Using this program it is
possible to calculate UV irradiance and UV-resources for different atmospheric conditions at a given geographic
location and specified time. Based on the threshold for vitamin D and erythemally active irradiance the UV
resources are obtained for various skin types and open body fraction. According to the classification we consider
different categories of UV-deficiency, UV-optimum and UV-excess (Chubarova and Zhdanova, 2013). The
interactive on-line UV tool represents a client-server application where the client part of the program is the web-
page with a special form for the input parameters required for erythemally-weighted UV irradiance simulations. The





server part of the program consists of the web-server and the CGI-script, where the different input parameters are set
by a user or taken from the climatological data available at the same site. In addition, in this part of the program
erythemally-weighted irradiance is calculated, visualized and classified according to the proposed UV resources
categories. The proposed UV irradiance altitude parameterization has been also incorporated in the calculation
scheme with additional account for the changes in the atmospheric parameters with the height. This enable a user to
evaluate UV irradiance at any requested elevation above sea level taking into account for a variety of the altitude
dependent parameters.
Let us analyze the UV resources for skin type 2 and the open body fraction of 0.25 in the Alpine region
(approximately 46°N and 7°E) for winter and spring noon conditions. On January, 15$^{th}$, the noon UV deficiency (no
vitamin D generation) conditions (with noon erythemally UV dose of about 97.2 Jm$^{-2}$) are observed at $H=0$ km for
typical (climatological) ozone, aerosol and surface albedo conditions, while at the same location the UV optimum is
observed at $H$ higher 0.5 km up to $H$=4.807 km (the highest point within the Alps, peak Mont Blanc) with noon
erythemally UV dose variation from 100.6 Jm$^{-2}$ to 122.9 Jm$^{-2}$. However, for skin type 4 according to the skin type
classification (Fitspatrick, 1988) the noon UV deficiency is observed at all altitudes and even at high surface albedo
$S$=0.9 corresponding to the pure snow with UV dose of 154.4 Jm$^{-2}$. If we increase the open body fraction for skin
type 4 from 0.25 to 0.5 the vitamin D generation is possible and, hence, the conditions are classified as UV
optimum.
On April, 15$^{th}$ for open body fraction 0.25 and typical climatological conditions at this geographical point the
moderate UV excess is observed for skin type 2, and the UV optimum - for skin type 4 at $H$=0 km with noon UV
dose of about 437.7 Jm$^{-2}$. At the altitude H = 2 km the conditions are characterized by the moderate UV excess for
skin type 2 and, still, by the UV optimum - for skin type 4 with UV dose of 463.4 Jm$^{-2}$. At the $H$=4 km a high UV
excess is observed for skin type 2 and the moderate UV excess - for skin type 4 with UV dose of 532.4 Jm$^{-2}$.
Thus, the proposed altitude UV parameterization can be effectively used for accurate estimating the BAUVR at
different altitudes with any altitude resolution for a variety of geophysical parameters over the PEEX domain in
Northern Eurasia. The accurate RT methods like Monte-Carlo, Discrete Ordinate method or others, of course, can be
used instead for UV irradiance simulations, however, their application is very time-consuming and are not possible
in some applications. The proposed approach is especially very useful for the application in different kinds of on-
line UV tools, where it is not possible to use a lot of prescribed calculations for a wide set of different geophysical
parameters or accurate UV modelling.
The combination of different altitude dependencies for main geophysical factors in the proposed parameterization
allows a user to make a reliable altitude UV assessment. It is not, of course, possible to take into account for the
specific altitude dependencies. We should also emphasize that the proposed ozone and aerosol altitude dependencies
in the troposphere were taken from the experimental data obtained in background conditions and, hence, should be
applied only for these conditions. However, they can be easily substituted using the proposed parameterization.
With the application of the proposed method we can also reveal the effects of different geophysical factors on
various types of BAUVR and estimate their comparative role in the altitude UV effects.  And, of course, it can be
also used in downscaling the UV results for the regions located at high altitudes from the coarse resolution global
chemistry-climate  models. The proposed method can be applied not only over the PEEX domain but on a global
scale over the world. However, more attention should be paid in this case to the evaluation of the particular altitude
dependence of the different parameters.





**5.Conclusions**
The objective of this paper was to develop a flexible parameterization based on rigorous model simulations with
account for generalized altitude dependencies of molecular density, ozone, and aerosols considering surface albedo
conditions. We show that for the specified groups of parameters we can present the altitude UV amplification ($A_{UV}$)
for different BAUVR as a composite of independent contributions of UV amplification from different factors with
the mean uncertainty of 1% and standard deviation of 3%. The parameterization takes into account for the altitude
dependence of molecular number density, ozone content, aerosol loading, and spatial surface albedo. We also
provide the generalized altitude dependencies of different parameters for evaluating the $A_{UV}$. Their validation against
the accurate RT model (8 stream DISORT RT code) for different types of BAUVR shows a good agreement with
maximum uncertainty of few percents and correlation coefficient $r>0.996$. It was not possible, of course, to cover
all the observed variety in the parameters. However, due to the proposed approach the parameter altitude
dependencies can be easily substituted by a user.
Using this parameterization one can estimate the role of different atmospheric factors in the altitude UV variation.
The decrease in molecular number density, especially at high altitudes, and the increase in surface albedo play the
most significant role in $A_{UV}$ growth. At high solar elevations the UV amplification due to aerosol at H=8.848 km
does not exceed 1.3 even when $AOD_{340}=0.8$ at $H$=0 km. The UV amplification due to aerosol calculated with the
Alpine-type AOD altitude aerosol dependence is much more pronounced than that calculated with the Asian-type
AOD altitude dependence, especially at relatively lower altitudes ($H$=2-4 km). The UV amplification due to ozone
does not exceed 1.20 and is higher at smaller solar elevations, especially, for vitamin-D-weighted irradiance.
This parameterization was applied to the on-line tool for calculating the UV resources (http:/momsu.ru/uv/) over the
PEEX domain. Using this tool one can easily evaluate the UV conditions (UV deficiency, UV optimum or UV
excess) at different altitudes for a given skin type and open body fraction. As an example, we analyzed the altitude
UV increase and its possible effects on health considering different skin types and various open body fraction for
January and April conditions in the Alpine region. We showed that even in clear sky conditions over the same
geographical point (46°N, 7°E) in mid-January the UV optimum can be observed higher H=0.5 km for skin type 2
while the UV deficiency are still remained at the altitudes up to H=4.8 km for skin type 4. In mid-April the account
for the altitude dependence at 4 km provides the changes from UV-optimum to UV excess for people with 4 skin
type, and from moderate UV excess to the high UV excess conditions -  for people with 2 skin type.
This approach can be also used in downscaling the results of global chemistry-climate models with the coarse spatial
resolution in mountainous domain and as a simple tool for different types of applications for personal purposes of
users.
**Acknowledgements**
The work was partially supported by the RFBR grant № 15-05-03612. We would like to thank all AERONET site
PI's which data were used for obtaining the aerosol altitude dependence. We also are grateful to the LIVAS team for
providing the aerosol extinction climatology.

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





**FIGURE CAPTIONS**

Fig.1. The comparison of $A_{UV}$ amplification factor calculated from Eq.(4) as multiplication of $A_M A_X A_{AOD} A_S$ with
the direct model simulation of UV amplification. All the parameters $(A_{UV}, A_M A_X A_{AOD} A_S)$ were obtained from
accurate model simulations.
Comment. The simulations were performed for different altitudes *(H=0 and H=5km)*, aerosol optical depth
*(AOD340= 0, 0.2, 0.4),* total ozone (*X*=250,300,350 DU), surface albedo (*S=0, S=0.9*) and solar elevations (h=20°
and 50°). For estimating the UV amplification we assume at H=0 km the conditions with 350DU, AOD340=0.4,
S=0% and normalized the BAUVR at the altitude H=5km to the value calculated with these parameters.
Fig.2. UV amplification due to decrease in molecular number density with the altitude *H* according to accurate
model simulations: TUV, 8- stream DISORT TUV model. *h*=50°. *X*=300 DU.
Fig.3. The altitude dependence of aerosol optical depth at 340nm with 1 sigma error bar according to the
AERONET, LIVAS and the Moscow State University datasets over European and Asian regions. May-September
period. The AOD at 330 nm the Moscow State University dataset and the AOD at 355nm from the LIVAS datasets
were recalculated to AOD at λ=340 nm using the Angstrom parameter α=1.3. See further details in the text.
Fig.4. UV amplification due to the surface albedo increase in mountainous areas according to different experimental
data and model simulations. The error bars of model simulation relates to the different input parameters – altitude of
2 and 3 km, solar elevation of 10, 30 and 50°, total ozone X=350DU, AOD340=0.17 at P=95%.
Fig. 5. The dependence of $r_{bio}$ with the altitude for different BAUVR from accurate model simulations for a variety
of geophysical parameters (left axis) and maximum $A_S$ effects due to changes in surface albedo from *S=0* at *H*=0
km to *S=1* at level *H* (right axis). The $r_{bio}$ regressions are shown in dashed line. Note, that the regression line for
$r_{Qeye}(H)$ is the same as for $r_{Qery}(H)$. The coefficients of the regression equations and the ranges of the input
parameters at H=0 are given in Table 2.
Fig.6. The comparison between the total altitude UV amplification according to the proposed method and the $A_{UV}$
values evaluated using the accurate RT model (TUV, 8-stream DISORT method).
Fig.7. The comparison between the simulated UV amplification according to the proposed parameterization and the
UV amplification from the experimental data as a function of altitude. Moscow State University dataset. Solar
elevation h=50°. Clear sky conditions. Note: since we used the data of different field campaigns the ozone altitude
gradient differed from the typical value. The total ozone was equal to X~300 DU at H=0km, X ~240 DU at H>3 km
and X ~250 DU at H~1-2 km.
Fig.8. Total UV amplification as a function of the altitude for different types of BAUVR in a variety of atmospheric
conditions with *S*=0 (a) and *S*=0.9 (b). The model parameters at H=0 km: *X*=250-350 DU, $AOD_{340}$=0.2-0.4. The
Alpine type of AOD altitude dependence according to the Eq. (15) was taken into account. Solar elevation h=50°.
Fig.9. The UV amplification due to molecular *$A_M(Qery)$, $A_M(QvitD)$,*ozone *$A_X(Q_{vitD})$, $A_X(Q_{ery})$,* aerosol $A_{AOD,f1(H)}$,
$A_{AODf2(H)}$for the Alpine *f1(H)* and Asian *f2(H)* types of altitude dependences, and surface albedo $A_S(Qery)$, $A_S(QvitD)$
changes with the altitude (a) and their total altitude effect on $A_{UV}$ for different types of BAUVR (b). At *H*=0 km:
$AOD_{340}$=0.8, *X*=250 DU. The surface albedo has an abrupt change at 2 km from S=0 to S=0.95. Solar elevation -
*h*=50°.





Table 1. Relative molecular gradients $G_{bio\_M(A=0)}$(%/km) at different solar elevations and ozone content for different
types of BAUVR. Accurate model simulations. Zero surface albedo conditions. Determination coefficient $R^2$ is
higher than 0.997 in all cases. The standard error of $G_{bio\_M(A=0)}$ is given in the brackets at P=95% .

| h, solar elevation, degrees | erythemally-weighted irradiance | cataract-weighted irradiance | vitamin D-weighted irradiance | erythemally-weighted irradiance | cataract-weighted irradiance | vitamin D-weighted irradiance |
|---|---|---|---|---|---|---|
| | X=300 DU | | | X=500 DU | | |
| 10 | 4.5 (0.04) | 3.8 (0.04) | 1.8 (0.01) | 4.8 (0.05) | 3.9 (0.04) | 0.8 (0.03) |
| 20 | 6.4 (0.04) | 6.9 (0.05) | 7.1 (0.06) | 6.0 (0.03) | 6.6 (0.04) | 6.8 (0.05) |
| 30 | 6.7 (0.01) | 7.2 (0.02) | 7.8 (0.02) | 6.1 (0.01) | 7.0 (0.01) | 7.8 (0.02) |
| 40 | 6.4 (0.02) | 6.8 (0.01) | 7.3 (0.01) | 5.8 (0.02) | 6.6 (0.02) | 7.4 (0.01) |
| 50 | 6.0 (0.03) | 6.2 (0.03) | 6.7 (0.03) | 5.5 (0.03) | 6.1 (0.03) | 6.8 (0.03) |
| 60 | 5.7 (0.04) | 5.8 (0.04) | 6.2 (0.04) | 5.3 (0.04) | 5.7 (0.04) | 6.4 (0.04) |



Table 2. The coefficients for calculating the $r_{bio}$ values (Eq. 21) for different types of BAUVR. Model estimations.
The standard error of the coefficients is given in the brackets at P=95%.

| | erythemally-weighted irradiance | cataract-weighted irradiance | vitamin D-weighted irradiance |
|---|---|---|---|
| $b$ | -0.025(0.0002) | -0.025(0.0002) | -0.025(0.0002) |
| $c$ | 0.394(0.0009) | 0.394(0.0009) | 0.405(0.0008) |
| $R^2$ | >0.99 | >0.99 | >0.99 |

Note: the simulations were fulfilled for different combinations of input parameters at $H=0$:
$X$=250-350 DU, $AOD_{340}$=0.2-0.4, $S$=0-0.9, $h$=20-50°.







**FIGURES**

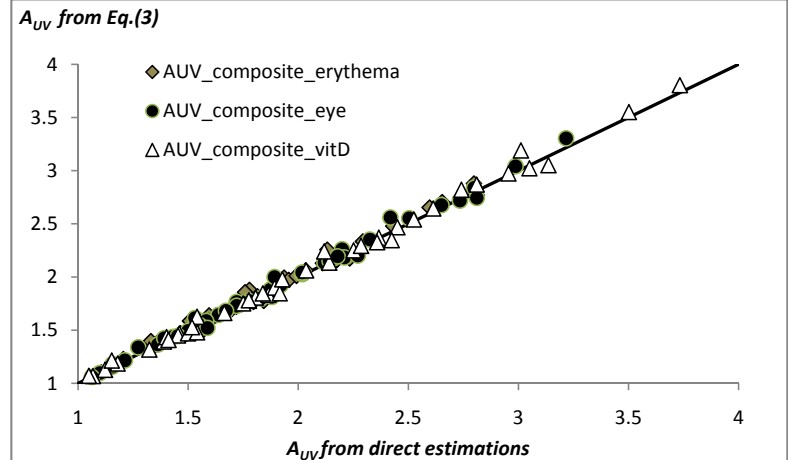


**Figure 1. The comparison of $A_{UV}$ amplification factor calculated from Eq.(4) as multiplication of $A_M A_X A_{AOD} A_S$ with the**
**direct model simulation of UV amplification. All the parameters ($A_{UV}$, $A_M A_X A_{AOD} A_S$) were obtained from accurate model**
**simulations.**
*Comment. The simulations were performed for different altitudes (H=0 and H=5km), aerosol optical depth (AOD340= 0, 0.2,*
*0.4, total ozone (X=250,300,350 DU), surface albedo (S=0, S=0.9) and solar elevations (h=20° and 50°). For estimating the*
*UV amplification we assume at H=0 km the conditions with 350DU, AOD340=0.4, S=0% and normalized the BAUVR at the*
*altitude H=5km to the value calculated with these parameters.*








**Figure 2. UV amplification due to decrease in molecular number density with the altitude $H$ according to accurate**
**model simulations: 8- stream DISORT TUV model. $h=50°$. $X=300$ DU. $R^2>0.997$ – for all regression lines.**






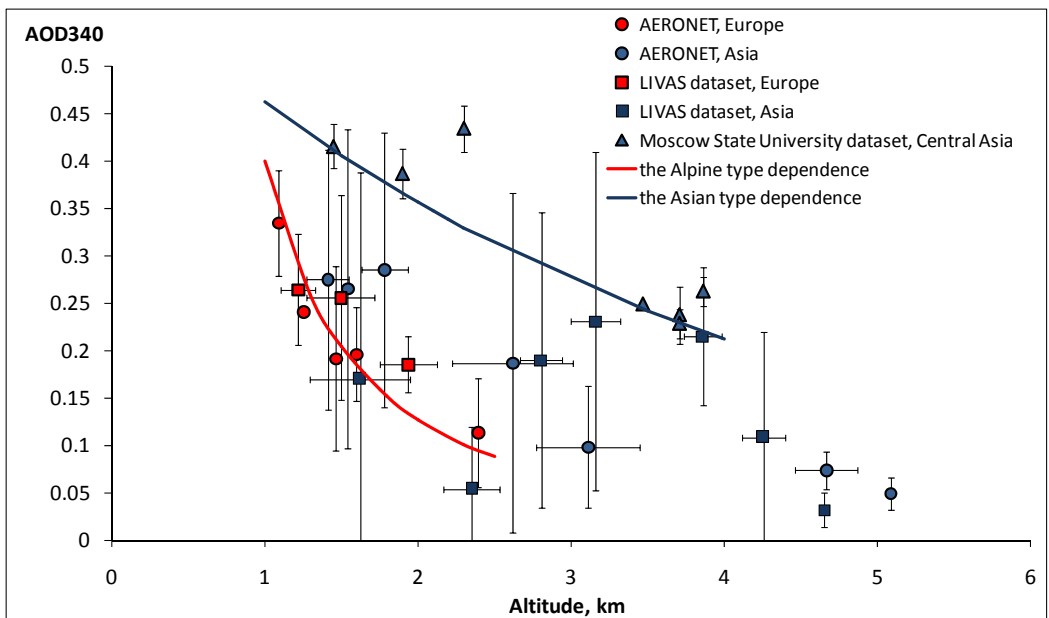


**Figure 3. The altitude dependence of aerosol optical depth at 340nm with 1 sigma error bar according to the AERONET,**
**LIVAS and the Moscow State University datasets over European and Asian regions. May-September period. The AOD at**
**330 nm the Moscow State University dataset and the AOD at 355nm from the LIVAS datasets were recalculated to AOD**
**at λ=340 nm using the Angstrom parameter α=1.3. See further details in the text.**




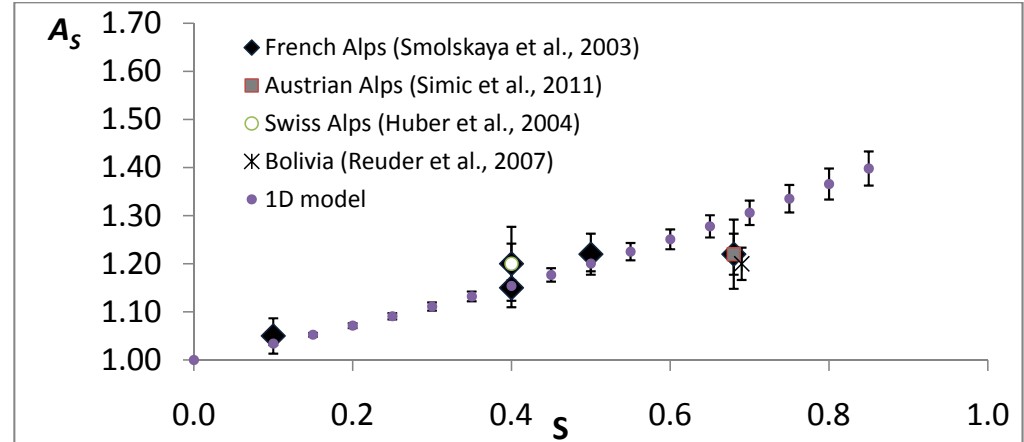


**Figure 4. UV amplification due to the surface albedo increase in mountainous areas according to different**
**experimental data and model simulations. The error bars of model simulation relates to the different input**
**parameters – altitude of 2 and 3 km, solar elevation of 10, 30 and 50°, total ozone X=350DU, AOD$_{340}$=0.17 at**
**P=95%.**




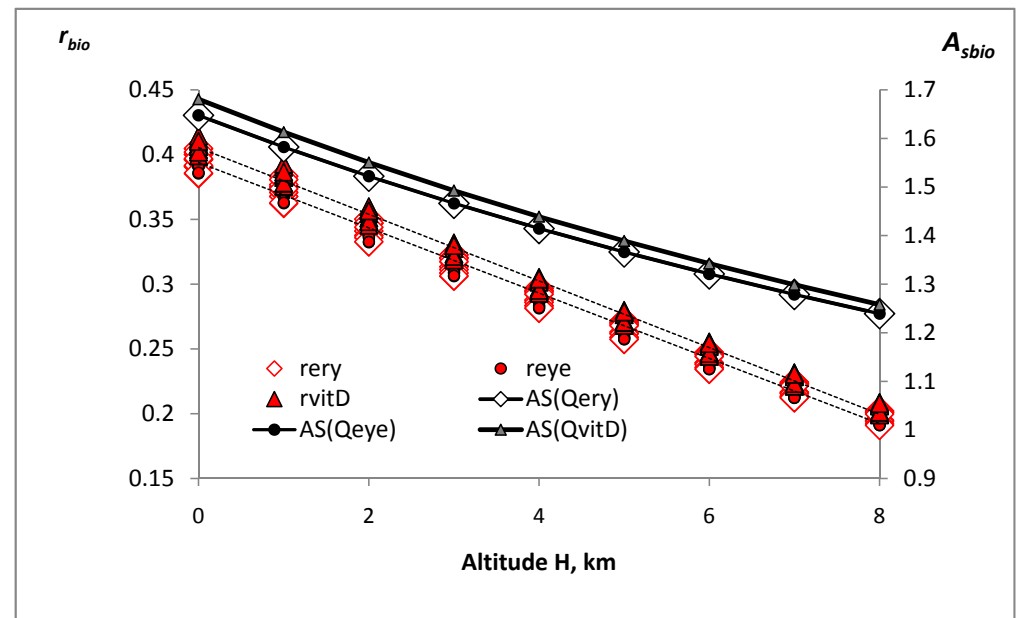

**Figure 5. The dependence of $r_{bio}$ with the altitude for different BAUVR from accurate model simulations for a**
**variety of geophysical parameters (left axis) and maximum $A_S$ effects due to changes in surface albedo from $S=0$ at**
**$H=0$ km to $S=1$ at level $H$ (right axis). The $r_{bio}$ regressions are shown in dashed line. Note, that the regression line for**
**$r_{Oeve}(H)$ is the same as for $r_{Oerv}$ $(H)$. The coefficients of the regression equations and the ranges of the input**
**parameters at $H=0$ are given in Table 2.**




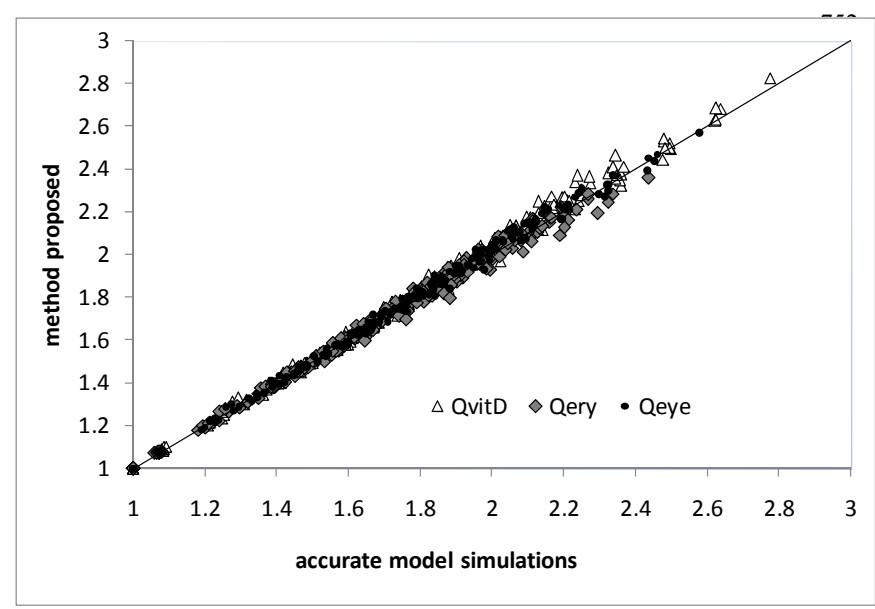

**Figure 6. The comparison between the total altitude UV amplification ($A_{UV}$) according to the proposed method and**
**the$A_{UV}$ values evaluated using the accurate RT model (TUV, 8-stream DISORT method).**







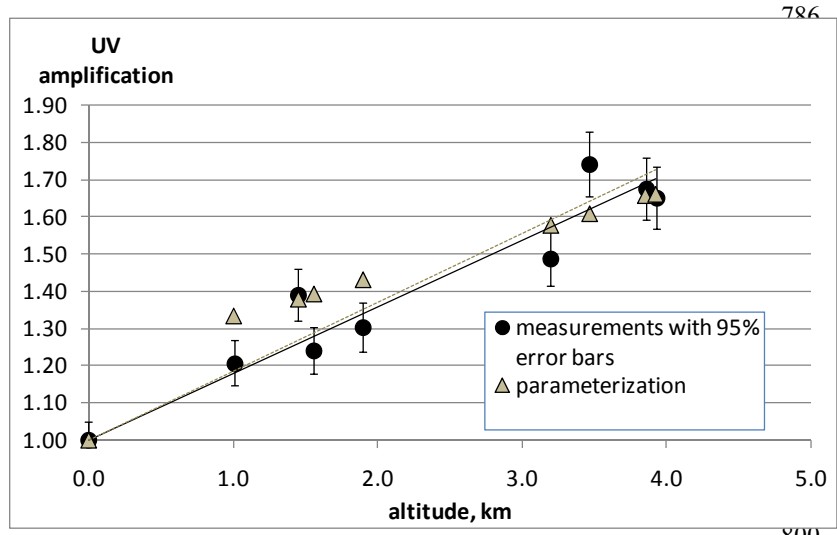





**Figure 7. The comparison between the simulated UV amplification according to the proposed parameterization and the UV amplification from the experimental data as a function of altitude. Moscow State University dataset. Solar elevation h=50°. Clear sky conditions. Note: since we used the data of different field campaigns the ozone altitude gradient differed from the typical value. The total ozone was equal to X~300 DU at H=0km, X ~240 DU at H>3 km and X ~250 DU at H~1-2 km.**

















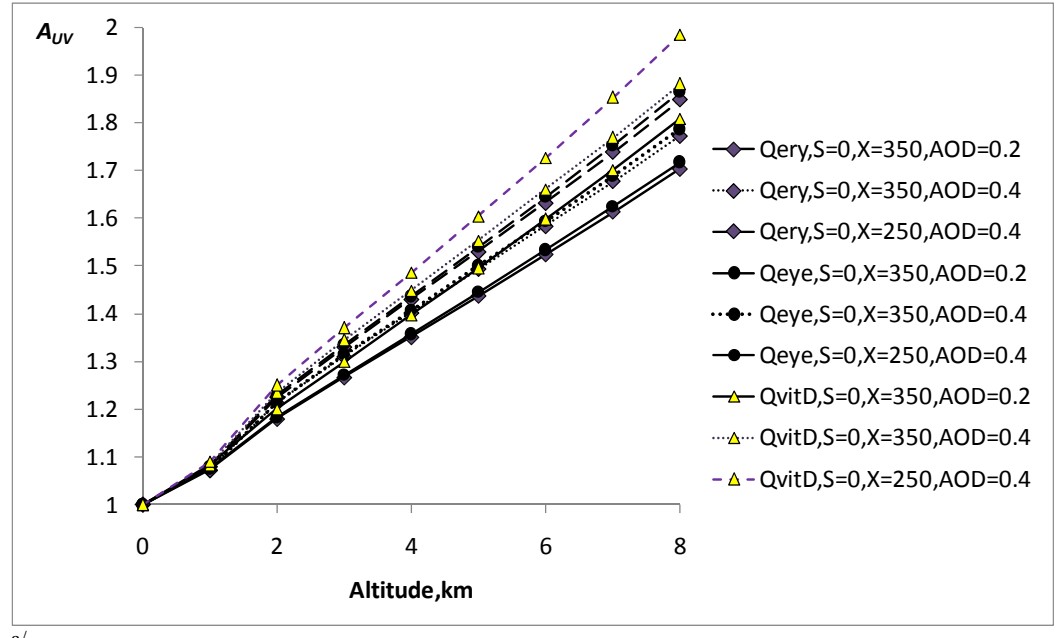

a/

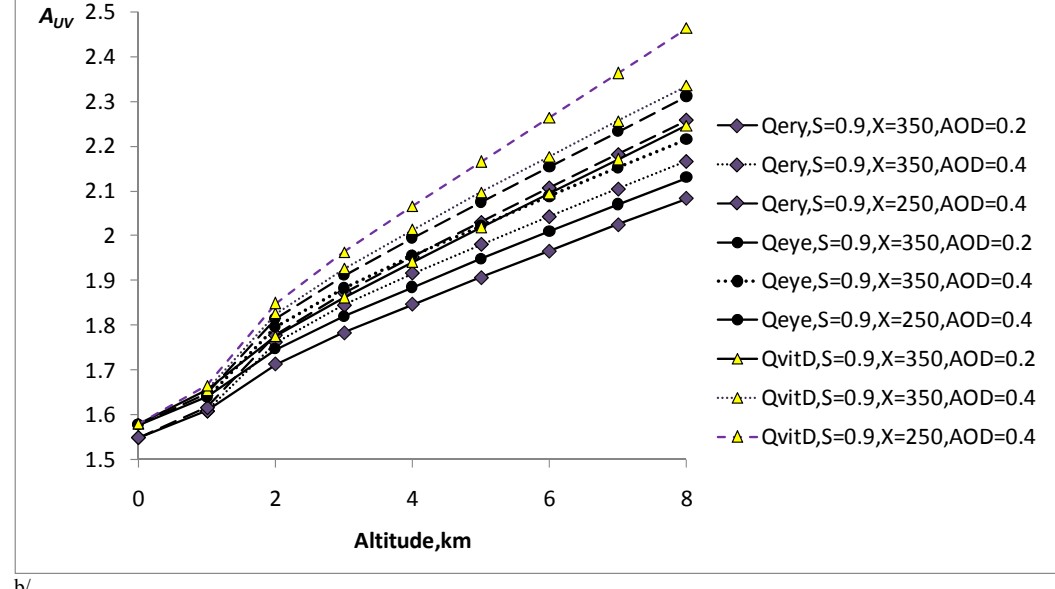

b/
**Figure 8. Total UV amplification as a function of the altitude for different types of BAUVR in a variety of**
**atmospheric conditions with *S*=0 (a) and *S*=0.9 (b). The model parameters at H=0 km: *X*=250-350 DU, *AOD*$_{340}$=0.2-**
**0.4. The Alpine type of AOD altitude dependence according to the Eq. (15) was taken into account. Solar elevation-**
**h=50°.**



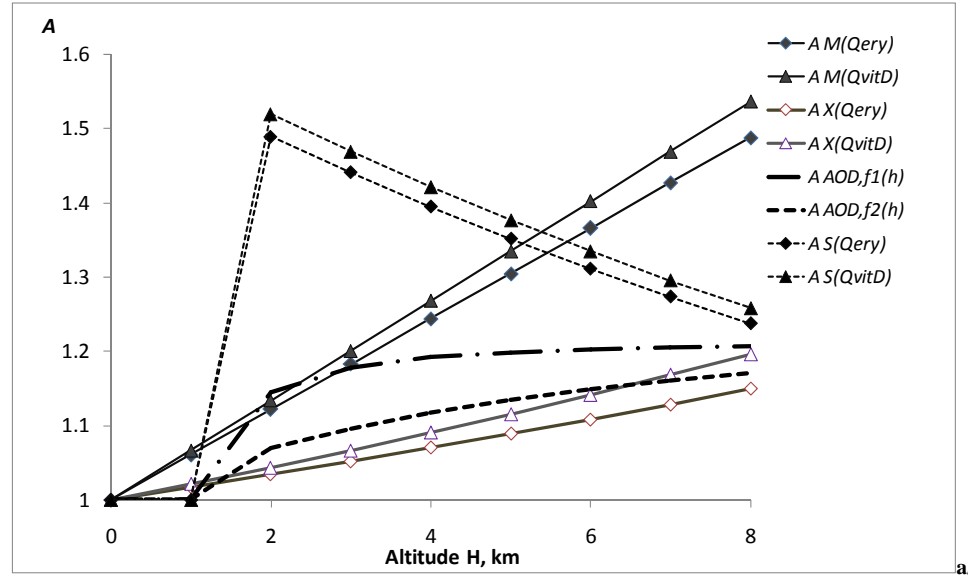

823                                                                                                    a/

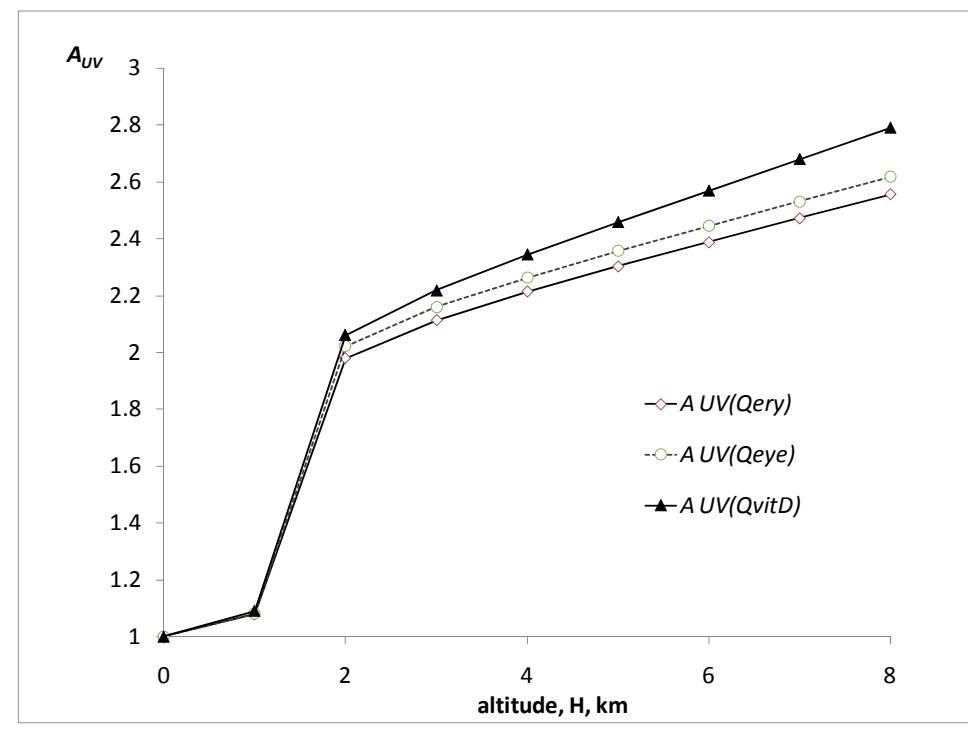

825                                                                                                    b/

**Figure 9. The UV amplification due to molecular $A_M(Qery)$, $A_M(QvitD)$, ozone $A_X(Q_{vitD})$, $A_X(Q_{ery})$, aerosol $A_{AOD,f1(H)}$,**
**$A_{AODf2(H)}$ for the Alpine $f1(H)$ and Asian $f2(H)$ types of altitude dependences, and surface albedo $A_S(Qery)$, $A_S(QvitD)$**
**changes with the altitude (a) and their total altitude effect on $A_{UV}$ for different types of BAUVR (b). At $H=0$ km:**
**$AOD_{340}=0.8$, $X=250$ DU. The surface albedo has an abrupt change at 2 km from $S=0$ to $S=0.95$. Solar elevation - $h=50°$.**