# Peer review of "A new parameterization of the UV irradiance altitude dependence for clear-sky conditions and its application in the on-line UV tool over Northern Eurasia"

_Atmospheric Chemistry and Physics, 2016_

## Referee Comment (RC1) · Anonymous Referee #2 · 28 May 2016

General comments:

The authors of the paper propose a parametrization of the altitude effect on three types of biologically effective UV irradiance. In such context this research thoroughly explores the amplification of the effective irradiances as a function of the altitude variation of molecular number density, of ozone and aerosol, and albedo. The implementation of the UV parametrization in the on-line UV tools can be of potential interest to the researchers involved in studies on the assessment of human UV exposures. The analysis is comprehensive and it will be acceptable for publication after taking into consideration

the issues underlined below.

Specific comments: Introduction L32:The predominant factors which interact with UV radiation determining its variability at the Earth' surface are mentioned, however UV radiation is also controlled by the variation in the cyclic Sun emittance: the 27-day cycle leads to variations less than 1% for $\lambda$ >250nm , 6-8% in the band 245-250nm; the 11-year sunspot cycle determines small changes in irradiance and influences the shortest extra-terrestrial wavelengths. The above factors should be also included (Ref. S., Madronich. The atmosphere and UV-B radiation at ground level. [book auth.] A.R. Young (Eds.) L.O. Bjorn. Environmental UV Photobiology. New York : s.n., 1993, pp. 1-39.) L43: How the UV index is calculated should be better specified for readers not familiar with this parameter as well as its reference (COST-713. Action UVB Forecasting. European Communities. Brussels : s.n., 2000). L47-49: the photobiological quantities (erythemally –weighted irradiance, erythemal doses) should be defined. L46-47: the following references could be acknowledged: Siani, G.R. Casale, H. Diémoz , G. Agnesod, M.G. Kimlin, C.A. Lang and A. Colosimo, Personal UV exposure in high albedo alpine sites, Atmos. Chem. Phys., 2008, 8, 3749–60; Casale G. R., A. M. Siani, H. Diémoz, G. Agnesod, A.V. Parisi, A. Colosimo (2015) Extreme UV Index and Solar Exposures at Plateau Rosà (3500 m a.s.l) in Valle d'Aosta Region, Italy, Science of the Total Environment 512–513 (2015) 622–630; L65. The following reference should be also acknowledged: Seckmeyer, G., Mayer, B., Bernhard, G., Erb, R., Albold, A., Jager, H., Stockwell, W.R.: New maximum UV irradiance levels observed in Central Europe, Atmos. Environ., 31(18), 2971-2976, 1997. Materials L89: It is worth pointing up that biological action spectra, although helpful to understand the biological reaction, do not express direct information on the possible combined effects of different wavelengths. Additivity for wavelength contributions has been documented for the erythema action spectrum, but not for the vitamin D action spectrum (ref. M. Norval, L. Björn, F. R. de Gruijl, Is the action spectrum for the UV-induced production of pre-vitamin D3 in human skin correct?, Photochem. Photobiol. Sci., 2010, 9, 11–17). L91-98: The weighting function (action spectrum) is generally normalized to unity at

the wavelength of maximal sensitivity, in case of erythemal and Vitamin D action spectra are both normalized at 298 nm. Perhaps figures showing the discrepancy among the action spectra could better explain the wavelength-dependent effectiveness of UV radiation in causing the specific reactions. L137-144: More clarity is necessary in this part of the text mainly in "noon UV deficiency and UV deficiency category". How is the vitamin D threshold determined? What are the values of "UV excess" In the context of biomedical radiation effects it should be highlighted that the skin orientation relative to the Sun and the geometry of the human body, could strongly modify the results limited to UV irradiance measurements on horizontal surface. In addition since the beneficial effect of UV radiation is related to the body area of exposed skin, the length of time to produce sufficient vitamin D decreases with the increase of the exposed body area for all phototypes (See for example McKenzie, R.L., J. B. Liley and L. O. Bjorn (2009) UV Radiation: Balancing Risks and Benefits. Photochem. Photobiol., 85, 88–98.). Yet, obesity and age should also be mentioned as influential factors in vitamin D production (see The Relationship between Ultraviolet Radiation Exposure and Vitamin D Status in Nutrients 2010, 2, 482-495; doi:10.3390/nu2050482). In this regard it would be reasonable at least to acknowledge the above issue, whether in the Introduction or in the Discussion. Results L214: The quadratic and linear terms of eqs 6-7-8 have very small coefficients with respect to the constant term. The authors should provide the physical meaning of these equations, for example: when h=0, does the RAF(erythema or vitamin D ) account the diffuse component? The units of the coefficients in eqs should be specified. Numbers of the coefficients in Eqs 8 and 9 should be expressed in the same form L286: The authors should give more details about " the coefficients have been re-affirmed using more statistics". In discussion or in conclusions: the authors should point out that their analysis is based on irradiance and the question of how well the radiation received by the anatomical area is related to that incident on a horizontal surface should be discussed. To determine the individual levels of UV exposure, that is, the real biologically effective doses of sunlight, dosimeters which have a spectral response almost identical to that of the UV-induced photobiological effect, should be

mentioned.

Minor comments: web addresses could be in a web reference list. L143: "Ccurrently " should be replaced by "Currently" Eq 11: the subscript "AOT-0" should be replaced by "AOT0"

---

## Referee Comment (RC2) · Anonymous Referee #1 · 31 May 2016

The paper presents a parametrization of the altitude effect on different types of biologically effective UV related irradiance. It is a useful tool for the scientific community using UVI measurements and it is related with specific health issues.

The analysis and the presentation of the results is adequate for publication in ACP after the authors take into account the following suggestions/comments.

Equations 7-9. What are the units used for the solar elevation here ? Since the coefficients are very small compared ot the constant factor. What is their physical meaning ?

Line 259. Since we expect the majority of aerosols to be found at lower altitudes, how realistic is to assume that the SSA is non altitude dependent. ?

SSA: there are publications for SSA at UV wavelengths (e.g. Arola et al., 2009 based Kinne et al simulations) http://onlinelibrary.wiley.com/doi/10.1029/2009GL041137/abstract, that report much lower values. Is the SSA=0.96 realistic for UV wavelengths?

Figure 3 : it would be easier for the reader if more colors could be used as for example aeronet Europe and Livas could be mixed now.

The provided uncertainty of 1% and 3% has to be clarified more. Here model inputs have errors as they come mostly from measurements. So if the authors would like to provide an uncertainty budget they have to include the propagation of errors coming from the actual measurements and/or fitting procedure they have used.

As an example LIVAS 0.1 difference from AERONET is not representative of the actual determination of the AOD at a certain height but as a total column AOD comparison among AERONET and LIVAS.

In addition, the abstract reads: "UV amplification from different factors within a wide range of their changes with mean uncertainty of 1% and standard deviation of 3% compared with the exact model simulations with the same input parameters. " It is not clear what the authors mean here.

L140-144: More about the threshold concerning vitamin D have to be reported.

---

## Referee Comment (RC3) · Anonymous Referee #3 · 31 May 2016

The manuscript deals with imports subject. i.e., modeling biologically effective UV (BEUV) radiations reaching the ground level. The authors provide useful formula for accurate estimation of the BEUV height dependence. It could improve various presently used UV index forecast models run by national weather services. Thus, the manuscript fits well with the journal scope. The reviewer decision is to accept the manuscript with minor changes listed below.

Minor comments:

l.168 – the cloud effects are mentioned here but all the calculations are carried out for

cloudless conditions. It was clearly stated by the authors. However, they recommend using the on line tool based on the proposed formula that could be used also for all-sky conditions. Thus they should comment on validity of this formula, especially "As" (albedo) term, for the partially cloudy conditions.

l.179. Angstrom exponent of 1.0 is proposed in the calculations. Authors found that "A" coefficients do not depend significantly on the aerosols characteristics. Probably we can use here any value of Angstrom exponent (the same concern other aerosols characteristics: ssa, asymmetry factor ) because A represents the relative value and the exponent value does not change with the altitude. Authors should comment on the selected aerosols values, which are proper for visible and UVA ranges, but not representative over UV-B range. For example, Angstrom exponent of 1 provides that AOD at 308 nm is about 10% larger than that at 340 nm but many authors suggested that Angstrom exponent in UVB range could be zero and even negative.

l.286." we can see its satisfactory agreement . . ..". I can not see the mentioned agreement. It is better to calculate the exponent value, separately for the AERONET and LIVAS data, and next discuss the agreement with the Pamir/Tien Shan exponent.

l. 348." The value r_bio has a relatively small dependence on altitude". Exactly, b value is small but c is also small. Figure 5 shows that r_bio changes significantly (0.4 for H=0 but 0.2 for H=8 km). Thus, for me it is not so small change.

l.425-439. The noon dose is mentioned many times but it is not clear how long is the exposition?, 1 hour around noon? Please provide 1 MED value and the vitamin D3 threshold dose for photo-type II and IV used in the calculations.

l.433. Open body fraction for skin phototype IV of 0.25 and 0.5 on 15th January is highly unreliable during the winter sunbathing. Value of 0.10 here is much more probable.

l. 718. Fig.3. Here Angstrom exponent=1.3 but 1 was used previously in the text (l.179, l.253).

---

## Author Comment (AC1) · 29 Jul 2016

Comments of the reviewer 1: The paper presents a parametrization of the altitude effect on different types of biologically effective UV related irradiance. It is a useful tool for the scientific community using UVI measurements and it is related with specific health issues.

The analysis and the presentation of the results is adequate for publication in ACP after the authors take into account the following suggestions/comments.

1. Equations 7-9. What are the units used for the solar elevation here? Since the

coefficients are very small compared ot the constant factor. What is their physical meaning? We used a simple polynomial regression method which provides better accounting the RAF dependence on solar elevation ( see their solar angle dependence in Figure below). The coefficients are small but they are very necessary. For example, for RAFQery at high solar elevations small coefficients at term 1 and term 2 are compensated by large multipliers (hˆ2 and h, respectively). For example, at h=90 first negative term gives -0.89, and the second term- 1.41, which are of the same order with the constant 0.665. Solar elevation is given in degree.

RAFQery(h)= -1.10E-04ïĆś1.49E-5 h2 + 1.57E-02ïĆś1.53E-3 h +0.665ïĆś0.0333

Since this is a statistical polynomial regression approach physical meaning of coefficients is not important as we should take into account all terms at once.The polynomial regression approach provides the most accurate account of the RAF dependence on h. We have added some clarification in the text about this standard method and some other details.

Figure. RAF dependence on solar elevation and polynomial regression for different types of BAUVR.

(Please, look at the Figures in the attached pdf file in the Supplement)

New variant of the text at line 226-227: "Using the results of accurate RT modelling and polynomial regression approach we have obtained RAF dependencies on solar elevation in degree over h=10-90ïĆř range for different types of BAUVR: "

2 Line 259. Since we expect the majority of aerosols to be found at lower altitudes, how realistic is to assume that the SSA is non altitude dependent. ?

We assume this is possible. According to aircraft measurements (Panchenko et al.,2012) SSA at 440nm (the closest wavelength to the UV) changed from 0.87 to 0.93 but this could be a specific feature observed over Western Siberia in visible spectrum for specific conditions. Unfortunately we found no information on SSA altitude

dependence in UV spectral region in typical conditions.

However, as mentioned in the text, due to extremely low AOD at high altitudes the UV effects to the SSA changes are negligible. We have added some additional comments to the text:

The updated text is the following at line 281-285: "In some conditions single scattering albedo and asymmetry factor for visible wavelengths may have the altitude dependence (see, for example, the results of aircraft measurements in Western Siberia (Panchenko et al., 2012)). However, there is no information on the altitude dependence of aerosol properties in UV spectral region from the in-situ measurements over the PEEX area. Note, that the uncertainty of neglecting the altitude changes in single scattering albedo significantly decreases at small AOD observed at high altitudes and only the altitude changes in aerosol optical depth are usually taken into account in the standard tropospheric aerosol models (WMO, 1986)."

3 SSA: there are publications for SSA at UV wavelengths (e.g. Arola et al., 2009 based Kinne et al simulations) http://onlinelibrary.wiley.com/doi/10.1029/2009GL041137/abstract, that report much lower values. We added the validation of the proposed parameterization over a wider range of the parameters (SSA=0.88, SSA=0.96, Angstrom exponent =0.6, and 1.5) and obtained the same results. We added a few additional references as well. The text and Figure 2 (new numbering) have been changed.

At line 184-189: "The model simulations were made for the altitude changes from zero to 5 km with the variations of aerosol optical depth at 340nm within AOD340ïĄ¿0.0-0.4, variations in total ozone from 350 to 250 DU, and surface albedo changes from zero to S=0.9 at different altitudes. As the input aerosol parameters, within UV spectral region we also used single scattering albedo SSA varying from 0.88 to 0.96, factor of asymmetry g=0.72, and Angstrom exponent ïĄą varying from 0.6 to 1.5, which are close to the aerosol background characteristics in Europe (Chubarova, 2009, Arola et

al., 2009)."

Figure 2. The comparison of AUV amplification factor calculated from Eq.(4) as multiplication of AM AX AAOD AS with the direct model simulation of UV amplification. All the parameters (AUV, AM AX AAOD AS) were obtained from accurate model simulations. Comment. The simulations were performed for different altitudes (H=0 and H=5km), aerosol optical depth (AOD340= 0, 0.2, 0.4), single scattering albedo (SSA=0.88, 0.96), Angstrom exponent ($\alpha$=0.6,1.0,1.5), total ozone (X=250,300,350 DU), surface albedo (S=0, S=0.9) and solar elevation (h=20ïĆř and 50ïĆř). For estimating the UV amplification we assume at H=0 km the conditions with 350DU, AOD340=0.4, S=0% and normalized the BAUVR at the altitude H=5km to the value calculated with these parameters.

Please, look at the Figures in the attached pdf file in the Supplement

In addition, we included the discussion on aerosol properties in UV –B and also added the references there.

At line 262-270: "The coefficients were obtained according to model simulations for 0<AOD340<0.8, single scattering albedo (0.8<SSA<1), airmass mïĄ¿sinh-1 (mïĆč 2), and Angstrom exponent ïĄąïĄ¿1 (0.6<ïĄą<1.5). Note, that these are typical changes in main aerosol properties for European conditions in UV-A spectral range (Chubarova, 2009). However, the Angstrom exponent in UV-B spectral region can differ from this range and be even negative in particular conditions depending on aerosol size distribution and optical properties (Bais et al., 2007). Single scattering albedo in UVB spectral range according to the results of different field campaigns (UNEP, 2015) may vary from 0.7 to 0.97 with low SSA in the presence of black and brown carbon aerosol. Some results demonstrates no existence of SSA spectral dependence in UV (Barnard et al., 2008, Arola et al., 2009) but some results shows its spectral character (UNEP, 2015)."

4 Is the SSA=0.96 realistic for UV wavelengths? We increased the range of SSA in validation and added the discussion in the text. The SSA=0.96 is the upper boundary

of the possible value. The validation of SSA=0.94 have demonstrated a satisfactory agreement with the UV-B measurements in Moscow (see Chubarova, 2009). However, anyone can use different SSA value within 0.8-1.0 range which is an independent parameter in equation (12).

Please, look at the updated text at line 184-189: "The model simulations were made for the altitude changes from zero to 5 km with the variations of aerosol optical depth at 340nm within AOD340ïA¿0.0-0.4, variations in total ozone from 350 to 250 DU, and surface albedo changes from zero to S=0.9 at different altitudes. As the input aerosol parameters, within UV spectral region we also used single scattering albedo SSA varying from 0.88 to 0.96, factor of asymmetry g=0.72, and Angstrom exponent ïĄą varying from 0.6 to 1.5, which are close to the aerosol background characteristics in Europe (Chubarova, 2009, Arola et al., 2009)."

5 Figure 3 : it would be easier for the reader if more colors could be used as for example aeronet Europe and Livas could be mixed now. The idea to use only two colors was to show by color the attribution to different regions: Europe and Asia domains. To distinguish between AERONET and LIVAS in Europe we decided to use different size of the markers (circles) to make the difference between these two datasets more clear.

Figure 4. The altitude dependence of aerosol optical depth at 340nm with 1 sigma error bar according to the AERONET, LIVAS and the Moscow State University datasets over European and Asian regions. May-September period. The AOD at 330 nm the Moscow State University dataset and the AOD at 355nm from the LIVAS datasets were recalculated to AOD at ïĄň=340 nm using the Angstrom exponent ïĄą=1.0. See further details in the text.

(Please, look at the Figures in the attached pdf file in the Supplement)

6 The provided uncertainty of 1% and 3% has to be clarified more. Here model inputs have errors as they come mostly from measurements. So if the authors would like to provide an uncertainty budget they have to include the propagation of errors

coming from the actual measurements and/or fitting procedure they have used. As an example LIVAS 0.1 difference from AERONET is not representative of the actual determination of the AOD at a certain height but as a total column AOD comparison among AERONET and LIVAS. We agree, that the uncertainty mainly is due to the errors in input parameters. But our aim was not to show the whole budget of uncertainties but just to compare the exact model simulations with the proposed parameterization ( fitting procedure) that should be made with the same parameters.

We have changed a little the text at line 193-196:

"The correlation for all BAUVR types is higher than 0.99 with the mean relative difference of -1±3% compared with the exact model simulations if the same input parameters are used. Hence, the proposed approach based on the independent account for the terms, which are affected by different geophysical factors can be applied with high accuracy."

We had included some assessment of the quality of LIVAS dataset because this is a quite new dataset and some comparison with widely known AERONET dataset may be useful for readers. Yes, we agree that "LIVAS 0.1 difference from AERONET is not representative of the actual determination of the AOD at a certain height but as a total column AOD comparison among AERONET and LIVAS". But we used total AOT from LIVAS dataset at the elevated altitudes. That is why we assume that we can leave the text as is.

7 In addition, the abstract reads: "UV amplification from different factors within a wide range of their changes with mean uncertainty of 1% and standard deviation of 3% compared with the exact model simulations with the same input parameters. " It is not clear what the authors mean here.

As it was discussed above we tried to show the uncertainty of the proposed approach - the altitude UV parameterization. To exclude the uncertainty due to the input parameters we have to use the same input parameters in simulations. That is why we have

added "compared with the exact model simulations with the same input parameters".

8 L140-144: More about the threshold concerning vitamin D have to be reported.

We added the additional information concerning the vitamin D threshold and some other details about the method used for UV resources determination. However, the detailed discussion and the full method description can be found in "Chubarova, N., Zhdanova, Ye.: Ultraviolet resources over Northern Eurasia, Journal of Photochemistry and Photobiology B: Biology, Elsevier, 127, 38-51, 2013."

The text is the following at line 140-155: "In addition, we estimated UV resources at different altitudes according to the approach given in Chubarova and Zhdanova (2013), which has been developed on the base of international classification of UV index (Vanichek et al., 2000) and the vitamin D threshold following the recommendations given in CIE (2006). In CIE (2006) there were simple recommendations of choosing the minimum vitamin D dose (MVitDD) threshold using one fifth Minimal Erythemal Dose (MED) for a one fifth body area. In this study according to the new guidelines a healthy level of vitamin D3 was increased from 400 IU recommended in CIE (2006) to 1000 IU (Rationalizing nomenclature for UV doses..., 2014). The possibility to account for the open body fraction as a function of the effective air temperature was also applied in the UV resources estimating method (Chubarova, Zhdanova 2013) as it had been proposed in (McKenzie et al.,2009). According to this approach we defined noon UV deficiency when UV dose is smaller than the vitamin D threshold during 11:30-12:30 noon period, and 100% UV deficiency category, when it is not possible to receive vitamin D throughout the whole day. The UV optimum category is determined when the UV dose does not exceed erythemal threshold but it is possible to receive UV dose, necessary for vitamin D at noon hour. Several subclasses of UV excess are attributed to the thresholds depending on the standard UV index categories: moderate UV excess class, which relates to moderate category of hourly UV index, high UV excess, very high UV excess, and extremely high UV excess category. Further details about this approach can be found in (Chubarova, Zhdanova, 2013)."

Response to the reviewer 2: General comments: The authors of the paper propose a parametrization of the altitude effect on three types of biologically effective UV irradiance. In such context this research thoroughly explores the amplification of the effective irradiances as a function of the altitude variation of molecular number density, of ozone and aerosol, and albedo. The implementation of the UV parametrization in the on-line UV tools can be of potential interest to the researchers involved in studies on the assessment of human UV exposures. The analysis is comprehensive and it will be acceptable for publication after taking into consideration the issues underlined below.

Specific comments: 1. Introduction L32:The predominant factors which interact with UV radiation determining its variability at the Earth' surface are mentioned, however UV radiation is also controlled by the variation in the cyclic Sun emittance: the 27-day cycle leads to variations less than 1% for $\lambda$ >250nm , 6-8% in the band 245-250nm; the 11-year sunspot cycle determines small changes in irradiance and influences the shortest extra-terrestrial wavelengths. The above factors should be also included (Ref. S., Madronich. The atmosphere and UV-B radiation at ground level. [book auth.] A.R. Young (Eds.) L.O. Bjorn. Environmental UV Photobiology. New York : s.n., 1993, pp. 1-39.)

We added the solar activity factor in the text and added the reference. However, we decided not to include the details since this is a minor factor for the wavelengths larger than 300nm. At line 33-35: UV radiation is affected by astronomical factors (solar zenith angle, solar-earth distance, solar activity), by different atmospheric characteristics (total ozone content, cloudiness, aerosol, optically-effective gases), and by surface albedo (Madronich, 1993, Bais et al., 2007, Bekki et al., 2011).

We added an additional reference: S., Madronich. The atmosphere and UV-B radiation at ground level. [book auth.] A.R. Young (Eds.) L.O. Bjorn. Environmental UV Photobiology. New York : s.n., 1993, pp. 1-39.

2. L43: How the UV index is calculated should be better specified for readers not familiar with this parameter as well as its reference (COST-713. Action UVB Forecasting. European Communities. Brussels : s.n., 2000).

We have made the necessary changes in the text. We added the definition in the footnote and included the reference to "K. Vanicek, T. Frei, Z. Litynska, A. Schmalwieser. UV-Index for the Public, COST-713 Action, Brussels, 2000, 27p" has been already included in the reference list.

Footnote 1: "UV index is a widely used characteristic which is equal to erythemally-weighted irradiance expressed in (W m-2) multiplied on 40 (Vanicek et al., 2003)"

L47-49: the photobiological quantities (erythemally –weighted irradiance, erythemal doses) should be defined. We made several corrections in the text and left the terms "erythemally-weighted irradiance" and UV index, which was defined by equation (1) and in the footnote 1 as it was shown above..

The text has been changed in the following way at line 48: "At the European alpine stations in summer conditions the UV indices are often higher than 11 (Hülsen, 2012). For example, high UV index up to 12 was observed in mountainous areas in Italy (Casale et al. 2015). A significant UV growth with the altitude was also obtained at different sites in Austria and Switzerland (Rieder et al., 2010). In winter, erythemally-weighted UV irradiance is about 60% higher than that at lower-altitude European sites (Gröbner et al., 2010)."

L46-47: the following references could be acknowledged: Siani, G.R. Casale, H. Diémoz , G. Agnesod, M.G. Kimlin, C.A. Lang and A. Colosimo, Personal UV exposure in high albedo alpine sites, Atmos. Chem. Phys., 2008, 8, 3749–60; Casale G. R., A. M. Siani, H. Diémoz, G. Agnesod, A.V. Parisi, A. Colosimo (2015) Extreme UV Index and Solar Exposures at Plateau Rosà (3500 m a.s.l) in Valle d'Aosta Region, Italy, Science of the Total Environment 512–513 (2015) 622–630;

Thank you for the useful references. We included one of them concerning the high UV index in the Introduction and another – in the Discussion section..

At line 49: For example, high UV index up to 12 was observed in mountainous areas in Italy (Casale et al. 2015).

At line 474 : In this case UV dosimeters which have a spectral response almost identical to that of the UV-induced photobiological effect (Siani et al., 2008) is the most accurate way for evaluating the individual levels of UV exposure.

L65. The following reference should be also acknowledged: Seckmeyer, G., Mayer, B., Bernhard, G., Erb, R.,Albold, A., Jager, H., Stockwell, W.R.: New maximum UV irradiance levels observed in Central Europe, Atmos. Environ., 31(18), 2971-2976, 1997.

Thank you. We also included this reference in the list.

At line 67: The accurate results of measurements from different field campaigns devoted to the evaluation of altitude UV effects shown in (Bernhard et al., 2008, Blumthaler and Ambach, 1988, Blumthaler et al., 1994, Blumthaler et al., 1997, Dahlback et al., 2007, Lenoble et al., 2004, Piacentini, et al. 2003, Pfeifer et al., 2006, Seckmeyer et al., 1997, Sola et al., 2008, Zaratti et al., 2003) provide precise, however, local character of this phenomenon, which results in various altitude UV gradients.

L89: It is worth pointing up that biological action spectra, although helpful to understand the biological reaction, do not express direct information on the possible combined effects of different wavelengths. Additivity for wavelength contributions has been documented for the erythema action spectrum, but not for the vitamin D action spectrum (ref. M.Norval, L. Björn, F. R. de Gruijl, Is the action spectrum for the UV-induced production of pre-vitamin D3 in human skin correct?, Photochem. Photobiol. Sci., 2010, 9, 11–17).

Yes, we agree and we added the discussion in the text on this point as a footnote 2.

[Figure]

At line 92: "We used erythemal action spectrum according to CIE (1998), vitamin D spectrum - according to CIE (2006)2, and cataract-weighted spectrum according to Oriowo et al. (2001)." The footnote 2: "Note, that a widely used conception of action spectra, which is based on the additivity of wavelength contribution, still has not be well documented for vitamin D action spectrum (Norval et al., 2010) and needs further studies."

L91-98: The weighting function (action spectrum) is generally normalized to unity at the wavelength of maximal sensitivity, in case of erythemal and Vitamin D action spectra are both normalized at 298 nm. Perhaps figures showing the discrepancy among the action spectra could better explain the wavelength-dependent effectiveness of UV radiation in causing the specific reactions.

Yes, we agree and we included the recommended Figure in the text.

At line 93: Various types of BAUVR action spectrum have different efficiency within the UV range (Fig.1). This Figure 1 is given below:

Figure 1. Action spectra for erythema (CIE, 1998), vitamin D (CIE, 2006) and for eye damage (cataract) (Oriowo et. al. 2001) effects. (Please, look at the Figures in the attached pdf file in the Supplement)

L137-144: More clarity is necessary in this part of the text mainly in "noon UV deficiency and UV deficiency category". How is the vitamin D threshold determined? What are the values of "UV excess"? We added some description of the method which was described in details in another paper.

The new variant is the following at line 140: "In addition, we estimated UV resources at different altitudes according to the approach given in Chubarova and Zhdanova (2013), which has been developed on the base of international classification of UV index (Vanichek et al., 2000) and the vitamin D threshold following the recommendations given in CIE (2006). In CIE (2006) there were simple recommendations of choosing the

minimum vitamin D dose (MVitDD) threshold using one fifth Minimal Erythemal Dose (MED) for a one fifth body area. In this study according to the new guidelines a healthy level of vitamin D3 was increased from 400 IU recommended in CIE (2006) to 1000 IU (Rationalizing nomenclature for UV doses…, 2014). The possibility to account for the open body fraction as a function of the effective air temperature was also applied in the UV resources estimating method (Chubarova, Zhdanova 2013) as it had been proposed in (McKenzie et al.,2009). According to this approach we defined noon UV deficiency when UV dose is smaller than the vitamin D threshold during 11:30-12:30 noon period, and 100% UV deficiency category, when it is not possible to receive vitamin D throughout the whole day. The UV optimum category is determined when the UV dose does not exceed erythemal threshold but it is possible to receive UV dose, necessary for vitamin D at noon hour. Several subclasses of UV excess are attributed to the thresholds depending on the standard UV index categories: moderate UV excess class, which relates to moderate category of hourly UV index, high UV excess, very high UV excess, and extremely high UV excess category. Further details about this approach can be found in (Chubarova, Zhdanova, 2013). "

" In the context of biomedical radiation effects it should be highlighted that the skin orientation relative to the Sun and the geometry of the human body, could strongly modify the results limited to UV irradiance measurements on horizontal surface.

We added this information in the "Discussion" section. At line 472: The current state of the online interactive tool does not take into account for the skin orientation relative to the Sun and the geometry of the human body which can modify the results limited to UV irradiance simulations on horizontal surface (Hess and Koepke, 2008, Vernez et al., 2014). In this case UV dosimeters which have a spectral response almost identical to that of the UV-induced photobiological effect (Siani et al., 2008) is the most accurate way for evaluating the individual levels of UV exposure. The vitamin D production can be also affected by other factors such as obesity and age (Engelsen, 2010). However, these are the tasks for the future work.

In addition since the beneficial effect of UV radiation is related to the body area of exposed skin, the length of time to produce sufficient vitamin D decreases with the increase of the exposed body area for all phototypes (See for example McKenzie, R.L., J. B. Liley and L. O. Bjorn (2009) UV Radiation: Balancing Risks and Benefits. Photochem. Photobiol., 85, 88–98.).

The open body fraction has been included as a parameter in the proposed interactive program. Please, look on the updated version of the text. At line 142: "In CIE (2006) there were simple recommendations of choosing the minimum vitamin D dose (MVitDD) threshold using one fifth Minimal Erythemal Dose (MED) for a one fifth body area. In this study according to the new guidelines a healthy level of vitamin D3 was increased from 400 IU recommended in CIE (2006) to 1000 IU (Rationalizing nomenclature for UV doses. . ., 2014). The possibility to account for the open body fraction as a function of the effective air temperature was also applied in the UV resources estimating method (Chubarova, Zhdanova 2013) as it had been proposed in (McKenzie et al.,2009)."

Yet, obesity and age should also be mentioned as influential factors in vitamin D production (see The Relationship between Ultraviolet Radiation Exposure and Vitamin D Status in Nutrients 2010, 2, 482-495; doi:10.3390/nu2050482).

We added the proposed items in the Discussion Section at line 476: "The vitamin D production can be also affected by other factors such as obesity and age (Engelsen, 2010). However, these are the tasks for the future work."

In this regard it would be reasonable at least to acknowledge the above issue, whether in the Introduction or in the Discussion. Results

We included all recommended changes and references in the Discussion Section. See the updated variants above.

L214: The quadratic and linear terms of eqs 6-7-8 have very small coefficients with

respect to the constant term. The authors should provide the physical meaning of these equations, for example: when h=0, does the RAF(erythema or vitamin D ) account the diffuse component?

We used the standard procedure with the standard polynomial regression parameterization. The equations can be used over the h=10-90ïČř range that is why we did not consider the conditions at h=0. Physical meaning of this equation is in the fact that RAF has the dependence on solar angle due to changes in spectral distribution of solar irradiance. Similar approach is given in many papers (see, for example, Herman et al., 2010). However, we proposed the equations with smaller number of terms and with high quality.

We showed above (in response to the first reviewer) the example that the small coefficients in the first and second term compared with the constant play an important role at high solar elevation providing the pronounced RAF dependence on solar elevation. Please, look an additional Figure and the text above.

The text has been clarified at line 226: "Using the results of accurate RT modelling and polynomial regression approach we have obtained RAF dependencies on solar elevation in degree over h=10-90ïČř range for different types of BAUVR: "

The units of the coefficients in eqs should be specified The units of the coefficients are different depending on the degree of the polynomial regression and the number of terms . They have a technical character and we guess that we do not need their unit specification.

We added the necessary information about the type of approximation in the text. Please, look at our response above

Numbers of the coefficients in Eqs 8 and 9 should be expressed in the same form Sorry.

Done:

RAFQery(h)= -1.10E-04ïĆś1.49E-5 h2 + 1.57E-02ïĆś1.53E-3 h +0.665ïĆś0.0333 (7) $R^2$ = 0.98 RAFQvitD(h)= 1.66E-4ïĆś1.0E-5 h2- 2.77E-2ïĆś 1.1E-3 h + 2.5121ïĆś0.0233 (8) $R^2$ = 0.997 RAFQeye(h)=1.43E-6ïĆś1.0E-6 h3– 2.02E-4ïĆś 6.6E-5 h2 + 4.83E-3ïĆś2.9E-3 h + 1.297ïĆś0.035 (9) $R^2$ =0.98

All fo=rmatted equations can be seen in a right form in the Supplement.

L286: The authors should give more details about " the coefficients have been re-affirmed using more statistics". We added some additional information. New variant at line 311:

" However, the coefficients have been re-affirmed according to the monthly mean AOD over 1999-2012 period (case number N=137)."

In discussion or in conclusions: the authors should point out that their analysis is based on irradiance and the question of how well the radiation received by the anatomical area is related to that incident on a horizontal surface should be discussed.

As we mentioned above we have added a discussion on this point at line 472: "The current state of the online interactive tool does not take into account for the skin orientation relative to the Sun and the geometry of the human body which can modify the results limited to UV irradiance simulations on horizontal surface (Hess and Koepke, 2008, Vernez et al., 2014). In this case UV dosimeters which have a spectral response almost identical to that of the UV-induced photobiological effect (Siani et al., 2008) is the most accurate way for evaluating the individual levels of UV exposure. The vitamin D production can be also affected by other factors such as obesity and age (Engelsen, 2010). However, these are the tasks for the future work."

To determine the individual levels of UV exposure, that is, the real biologically effective doses of sunlight, dosimeters which have a spectral response almost identical to that of the UV-induced photobiological effect, should be mentioned.

We added the text. Please, look on our response to the previous comment.

Minor comments: web addresses could be in a web reference list.

In this variant we did not add it to the reference list since this is not a reference but a part of the text. But if necessary iot could be easily done.

L143: "Ccurrently "should be replaced by "Currently" Done

Eq 11: the subscript "AOT-0" should be replaced by "AOT0" Unfortunately I was not able to find this misprint in eq 11. We used AOD throughout the text after the recommendation before publishing in ACPD.

Response to the reviewer 3: The manuscript deals with imports subject. i.e., modeling biologically effective UV (BEUV) radiations reaching the ground level. The authors provide useful formula for accurate estimation of the BEUV height dependence. It could improve various presently used UV index forecast models run by national weather services. Thus, the manuscript fits well with the journal scope. The reviewer decision is to accept the manuscript with minor changes listed below.

Minor comments: 1. Cloudless conditions. It was clearly stated by the authors. However, they recommend using the on line tool based on the proposed formula that could be used also for allsky conditions. Thus they should comment on validity of this formula, especially "As" (albedo) term, for the partially cloudy conditions.

We decided to remove this part of calculations at high altitudes in cloudy conditions for non zero surface albedo in the interactive tool until the updated version of the algorithm for cloudy conditions is ready. Now, when a user tries to choose the cloudy conditions at high altitudes the following comment appears: "Sorry, this part is under development. You could use the preliminary UV assessment for clear sky conditions at the specified altitude". We decided not to include any comments in the text not to confuse a reader.

2 l.179. Angstrom exponent of 1.0 is proposed in the calculations. Authors found that "A" coefficients do not depend significantly on the aerosols characteristics. Probably we can use here any value of Angstrom exponent (the same concern other aerosols

characteristics: ssa, asymmetry factor ) because A represents the relative value and the exponent value does not change with the altitude. Authors should comment on the selected aerosols values, which are proper for visible and UVA ranges, but not representative over UV-B range. For example, Angstrom exponent of 1 provides that AOD at 308 nm is about 10% larger than that at 340 nm but many authors suggested that Angstrom exponent in UVB range could be zero and even negative.

We have added the range of single scattering albedo and Angstrom exponent in validation. The updated Figure 2 (new numbering) has been done. We obtained the same statistics (correlation coefficient, averages, standard deviation, etc). Model simulations revealed that the changes within $\alpha$=0.6-1.5 provide plus- minus 1% difference for various types of BAUVR. Of course, in some conditions it can be abnormal negative change in Angstrom but this would be for specific conditions. In addition we should note that it is very hard to evaluate Angstrom exponent in UV-B region due to different gas absorption which can dramatically influence the evaluation of the Angstrom exponent ( see the discussion, for example, in referenced publication Chubarova et al., 2016. We have added the discussion on this account.

New text at line 184: "The model simulations were made for the altitude changes from zero to 5 km with the variations of aerosol optical depth at 340nm within AOD340ïA¿0.0-0.4, variations in total ozone from 350 to 250 DU, and surface albedo changes from zero to S=0.9 at different altitudes. As the input aerosol parameters, within UV spectral region we also used single scattering albedo SSA varying from 0.88 to 0.96, factor of asymmetry g=0.72, and Angstrom exponent ïĄą varying from 0.6 to 1.5, which are close to the aerosol background characteristics in Europe (Chubarova, 2009, Arola et al., 2009). We compared the AUV values calculated as a multiplication composite of different separate parameters (AM, AX, AAOD, and AS) according to Eq.(4) with the AUV values, which were estimated as a ratio of direct simulations of BAUVR at the altitude H=5 km and at zero ground level. The results of the comparisons are shown in Fig.2. One can see a good agreement between the AUV values obtained

using multiplication of AM AX AAOD AS and the AUV values from direct estimations of BAUVR. The correlation for all BAUVR types is higher than 0.99 with the mean relative difference of -1±3% compared with the exact model simulations if the same input parameters are used. Hence, the proposed approach based on the independent account for the terms, which are affected by different geophysical factors can be applied with high accuracy.

However, we understand that it is important to include the additional discussion on UV aerosol properties in the text. We included the additional discussion on the aerosol properties in UV-B region in 3.4. "Aerosol UV amplification with the altitude" and increase the application of the range of Angstrom exponent from 0.6 to 1.5.

New text at line 262: "The coefficients were obtained according to model simulations for 0<AOD340<0.8, single scattering albedo (0.8<SSA<1), airmass mïA¿sinh-1 (mïĆč 2), and Angstrom exponent ïĄąïĄ¿1 (0.6<ïĄą<1.5). Note, that these are typical changes in main aerosol properties for European conditions in UV-A spectral range (Chubarova, 2009). However, the Angstrom exponent in UV-B spectral region can differ from this range and be even negative in particular conditions depending on aerosol size distribution and optical properties (Bais et al., 2007). Single scattering albedo in UVB spectral range according to the results of different field campaigns (UNEP, 2015) may vary from 0.7 to 0.97 with low SSA in the presence of black and brown carbon aerosol. Some results demonstrates no existence of SSA spectral dependence in UV (Barnard et al., 2008, Arola et al., 2009) but some results shows its spectral character (UNEP, 2015). We should also note that direct evaluation of SSA in UV-B spectral region is difficult because UV attenuation due to aerosol can occur together with the absorption in this spectral region by different gases (ozone, sulphur dioxide, formaldehyde, nitrogen dioxide, etc.). As a result we used the aerosol properties at 340nm as input parameters for the BAUVR simulations since we consider typical aerosol conditions without forest fires and heavy industrial smoke. Radiative effects of the existing AOD spectral dependence are relatively small within the UV-B spectral range therefore we consider the

same coefficients for different types of BAUVR."

3 l.286." we can see its satisfactory agreement . . ..". I can not see the mentioned agreement. It is better to calculate the exponent value, separately for the AERONET and LIVAS data, and next discuss the agreement with the Pamir/Tien Shan exponent.

Sorry, we agree that the formulation is not exact and need to be changed. The idea was not to show the quantitative agreement between the different datasets but to reveal the existence of sharp and flat aerosol altitude dependence for so-called Alpine and Asian types and to analyze their effects on UV. As it was already mentioned in the text (old line numbers 300-301): "We should note that the proposed altitude AOD dependencies according to (15) and (16) are considered only as a first proxy for the most sharp and flat altitude dependencies."

The text was changed in the following way at line 303:

"The second one is characterized by a much more gradual altitude AOD340 decrease observed over flat elevated Asian regions according to the AERONET and LIVAS dataset and the data obtained during Moscow State University field campaigns at the high-altitude plateau at Pamir and Tyan'-Shan' mountainous regions in Central Asia. The main reason of such a character is the existence of the additional aerosol emission sources (i.e. loess, mineral aerosol) from the vast areas of deserts and semi-deserts elevated over sea level of up to 3-4 kilometers."

Instead of "The second one is characterized by a much more gradual altitude AOD340 decrease observed over flat elevated Asian regions. The main reason of such a character is the existence of the additional aerosol emission sources (i.e. loess, mineral aerosol) from the vast areas of deserts and semi-deserts elevated over sea level of up to 3-4 kilometers. In addition, Fig.3 demonstrates the AOD340 dependence on altitude according to the data obtained during Moscow State University field campaigns at the high-altitude plateau at Pamir and Tyan'-Shan' mountainous regions in Central Asia. We can see its satisfactory agreement with the second type of fAOD(H) obtained from

the AERONET and LIVAS dataset. "

l. 348." The value r_bio has a relatively small dependence on altitude". Exactly, b value is small but c is also small. Figure 5 shows that r_bio changes significantly (0.4 for H=0 but 0.2 for H=8 km). Thus, for me it is not so small change.

Sorry, we agree that this part of the text needs editing. This is not exact formulation. We have changed the text in the following way at line 369:

"According to the model estimations the value rbio in clear sky conditions has the dependence on altitude, which appears due to a decrease mainly in molecular and aerosol loading and can be parameterized by a simple regression as follows:"

l.425-439. The noon dose is mentioned many times but it is not clear how long is the exposition?, 1 hour around noon? Please provide 1 MED value and the vitamin D3 threshold dose for photo-type II and IV used in the calculations.

We added the clarification of the UV resources evaluation in the "Materials and methods" Section at line 142: "In CIE (2006) there were simple recommendations of choosing the minimum vitamin D dose (MVitDD) threshold using one fifth Minimal Erythemal Dose (MED) for a one fifth body area. In this study according to the new guidelines a healthy level of vitamin D3 was increased from 400 IU recommended in CIE (2006) to 1000 IU (Rationalizing nomenclature for UV doses..., 2014). The possibility to account for the open body fraction as a function of the effective air temperature was also applied in the UV resources estimating method (Chubarova, Zhdanova 2013) as it had been proposed in (McKenzie et al.,2009). According to this approach we defined noon UV deficiency when UV dose is smaller than the vitamin D threshold during 11:30-12:30 noon period, and 100% UV deficiency category, when it is not possible to receive vitamin D throughout the whole day. The UV optimum category is determined when the UV dose does not exceed erythemal threshold but it is possible to receive UV dose, necessary for vitamin D at noon hour. Several subclasses of UV excess are attributed to the thresholds depending on the standard UV index categories: moderate UV excess class, which relates to moderate category of hourly UV index, high UV excess, very high UV excess, and extremely high UV excess category. Further details about this approach can be found in (Chubarova, Zhdanova, 2013)."

We have also changed the following text ( line 425-439 previous numeration) at line 447: "Let us analyze the UV resources for skin type 2 and the open body fraction of 0.25 in the Alpine region (approximately 46ïĆřN and 7ïĆřE) for winter and spring noon conditions. For these conditions the vitamin D threshold is equal to 100 J/m2 and Minimal Erythemal Dose -250 J/m2. According to our estimates on January, 15th, at H=0 km for typical (climatological) ozone, aerosol and surface albedo conditions the noon UV deficiency (no vitamin D generation) is observed with noon erythemally UV dose of about 97.2 Jm-2, while at the same coordinates at H higher 0.5 km up to H=4.8 km (the highest point within the Alps, peak Mont Blanc) we obtain the UV optimum conditions with noon erythemal UV dose varying from 100.6 Jm-2 to 122.9 Jm-2. However, for skin type 4 (Fitspatrick, 1988) with vitamin D threshold of 180 J/m2 the noon UV deficiency is observed at all altitudes and even at high surface albedo S=0.9 corresponding to the pure snow with UV dose of 154.4 Jm-2. The decrease in open body fraction for this skin type from 0.25 to 0.05, which could take place in frosty weather, provides 100% UV deficiency, when no vitamin D can be generated during the whole day at the H=4.8 km and S=0.9. On April, 15th , at H=0 km and typical climatological conditions at this geographical point noon UV dose is about 437.7 Jm-2. This means that for the open body fraction of 0.4 the moderate UV excess is observed for skin type 2 and the UV optimum – for skin type 4 , when vitamin D threshold is 112.5 Jm-2 and MED threshold - 450 Jm-2. At the altitude H = 2 km the conditions are characterized by the moderate UV excess for both skin types 2 and 4 with UV dose of 463.4 Jm-2. At the H=4 km a high UV excess is observed for skin type 2 and the moderate UV excess - for skin type 4 with UV dose of 532.4 Jm-2." "

l.433. Open body fraction for skin phototype IV of 0.25 and 0.5 on 15th January is highly unreliable during the winter sunbathing. Value of 0.10 here is much more probable.

[Figure]

Yes. We agree. Since it was interesting to show the change in the class of UV resources, in the new variant we decreased the open body fraction to 0.05. The new variant of the text at line 456: "The decrease in open body fraction for this skin type from 0.25 to 0.05, which could take place in frosty weather, provides 100% UV deficiency, when no vitamin D can be generated during the whole day at the H=4.8 km and S=0.9."

l. 718. Fig.3. Here Angstrom exponent=1.3 but 1 was used previously in the text (l.179, l.253). Sorry, this is a misprint.

New variant ( the numbering has been changed from 3 to 4 due to including a new Figure 1): "Fig.4. The altitude dependence of aerosol optical depth at 340nm with 1 sigma error bar according to the AERONET, LIVAS and the Moscow State University datasets over European and Asian regions. May-September period. The AOD at 330 nm the Moscow State University dataset and the AOD at 355nm from the LIVAS datasets were recalculated to AOD at ïĄň=340 nm using the Angstrom exponent ïĄą=1.0. See further details in the text. "

Please also note the supplement to this comment:
http://www.atmos-chem-phys-discuss.net/acp-2016-210/acp-2016-210-AC1-supplement.pdf

―――――――――――――――

---

## Author Response (AR2)

**The response to the technical comments from Prof. Co-Editor Veli-Matti Kerminen (18 Aug 2016)**

The authors have addressed the referee comments. There are a few technical issues that should be taken care of before uploding the final files for publication:

*1) please replace "-" with "it is" in two places on line 101.*

Done
New variant:

"According to our estimates, for example, at high solar elevation (h=60°) and for the variety of other parameters (total ozone, aerosol and surface albedo) the effective wavelength for erythemally-weighted irradiance ($Q_{ery}$) is ~317 nm, for cataract-weighted irradiance ($Q_{eye}$) it is ~313 nm, and for vitamin D-weighted irradiance ($Q_{vitD}$) it is ~308 nm."

*2) strange character between 1.0 and 3.9 km on line 117*

Done
New variant:

"In addition, the dataset of historical Moscow State University complex field campaigns over mountainous areas at Pamir (38 - 40.5° N, 73 - 74° E, H=1.0 - 3.9 km), and Tyan'Shan' (43° N, 77° E, H=3.47 km) was applied in the analysis (Belinski et al., 1968)."

*3) what does ... in "UV doses…, 2014)"on line 146 mean?*
This means the reference. We have significantly changed the reference list and the reference now is given in a right ( understandable) way.

New variant:

"In this study according to the new guidelines a healthy level of vitamin D3 was increased from 400 IU recommended in Bouillon et al. (2006) to 1000 IU in McKenzie et al. (2014)."

*4) there should be a space between a quantity and unit in the text, as well as between different components of units (like J m-2, not Jm-2)*

We have made necessary changes throughout the text.

*5) The format of citing papers in the text should be correct throughout the text.*

We have changed the reference list and citing papers in the text according to the ACP guidelines.

*6) The reference list contain full journal names as well as journal names the are abreviations. Please be consistent and follow ACP guidelines.*

We have changed the reference list according to the ACP guidelines.